# The N-terminal PA domains of signal-peptide-peptidase-like 2 (SPPL2) proteases impact on TNFα cleavage
Christine Schlosser [1], Kinda Sharrouf [1], Alkmini A. Papadopoulou [1], Martina Haug-Kröper[1], Suman Singh[1], Maximilian Johler[1], Jonas Pettinger[1], Henrike Horn[2], Marco Koch[2,3], Sabine Hoeppner [1,5] ✉ & Regina Fluhrer [1,3,4,5] ✉

Signal peptide peptidase-like (SPPL) proteases, members of the intramembrane aspartyl protease family, have attracted increased interest due to their involvement in immune cell differentiation and cellular glycan structure regulation. However, the enzymatic domain involved in substrate recognition remains enigmatic. Here we provide evidence that the N-terminal protease-associated (PA) domains of the SPPL2 subfamily are involved in substrate recognition and discrimination of substrates that differ slightly in their luminal/extracellular domain. Presence of the SPPL2c PA domain impairs SPPL2a/b mediated tumor necrosis factor α (TNFα) initial cleavage, kinetics, and processivity in cells and in vitro. In contrast, the SPPL2a PA domain enhances processing by SPPL2b. Additionally, we demonstrate non-canonical shedding activity of SPPL3 on full-length TNFα and that the ability for consecutive cleavage differs within the SPPL-family and is mainly based on the SPPL2a/b membrane spanning body. This provides the basis to finally understand the mechanistic differences of these homologous proteases.

Signal peptide peptidase-like (SPPL) proteases are intramembrane proteases that hydrolyse peptide bonds within the transmembrane (TM) domain of type II membrane proteins, with their N-termini facing the cytosol ($N_{in}$)[1]. Together with signal peptide peptidase (SPP; EC 3.4.23.B24), the founding member of the SPP/SPPL family, and the presenilins, they constitute the family of mammalian intramembrane aspartyl proteases (EC 3.4.23)[2–4]. These proteases are all multipass TM proteins, which comprise 9 TM domains and share two conserved catalytic aspartyl residues embedded in a YD-motif in TM domain 6 and a GxGD-motif in TM domain 7[5]. Mutation of either aspartyl residue to, for instance, an alanine (D/A) results in catalytic inactivity of the respective protease[6–9]. Mutations in a conserved PAL motive in TM domain 9 also reduce protease activity or even inactivate the proteases[10]. Experimentally determined 3D structures of presenilins indicate that TM domain 9 is located close to the active site[11]. In mammals, four SPPL proteases are known: SPPL2a, SPPL2b, SPPL2c and SPPL3[2–4]. Among them, SPPL3 is the smallest but most conserved member, with human and murine proteins being identical[12]. It is non-glycosylated and localizes to the medial/early-trans-Golgi[13], where it cleaves various glycan-modifying enzymes resulting in the release of their catalytic domains and,

thus, controlling the equilibrium of glycan modifications on secretory and membrane proteins[14,15]. SPPL2a is a lysosomal protease, which contains a canonical tyrosine-based sorting motif of the YXXø type, which is sufficient for its localization to the lysosomal/late endosomal compartments[16]. By processing CD74, the invariant chain of the MHCII complex, SPPL2a impacts on the maturation of B lymphocytes and of certain dendritic cell (DC) subsets in mice and humans[17–21]. Although SPPL2b is mainly detected on the plasma membrane, SPPL2a and SPPL2b display some redundant functions[22,23]. For instance, the lectin-like oxidised lipoprotein receptor 1 (LOX-1) N-terminal fragment is substrate to both proteases and non-cleaved LOX-1 NTFs cause enhanced proatherogenic and pro-fibrotic signalling[24]. In addition, both proteases impact on recognition of fungal pathogens by cleaving dectin-1, a pattern recognition receptor (PRR)[25], and affect subcellular trafficking by mediating turnover of certain SNARE proteins[26]. While SPPL2a, SPPL2b and SPPL3 are expressed almost ubiquitously in human and mouse tissues[1,4], SPPL2c expression appears to be restricted only to developing male germ cells, where it is implicated in the formation of the sperm acrosome and the regulation of sperm motility[27,28]. Like SPP, SPPL2c is a protein of the endoplasmic reticulum (ER)[8,28],

[1]Biochemistry and Molecular Biology, Institute of Theoretical Medicine, Faculty of Medicine, University of Augsburg, Augsburg, Germany. [2]Anatomy and Cell Biology, Institute of Theoretical Medicine, Faculty of Medicine, University of Augsburg, Augsburg, Germany. [3]University of Augsburg, Center for Interdisciplinary Health Research, Augsburg, Germany. [4]Centre for Advanced Analytics and Predictive Sciences (CAAPS), University of Augsburg, Augsburg, Germany. [5]These authors jointly supervised this work: Sabine Hoeppner, Regina Fluhrer. ✉e-mail: sabine.hoeppner@med.uni-augsburg.de; regina.fluhrer@med.uni-augsburg.de

however, the signals and mechanisms that cause ER retention of SPPL2c remain enigmatic.

Although in the recent years the substrate spectra of SPPL proteases have grown steadily, the number of so far identified substrates remains relatively small compared to presenilins[22]. One reason for this is an incomplete understanding of how these proteases select their substrates. In general, efficient substrate recognition by a protease, requires specific determinants within the substrate protein, which are recognized by certain parts of the enzyme. While common determinants that qualify a substrate for cleavage by intramembrane aspartyl proteases have been subject to intensive investigation in the past years[22,29], the characteristics of the proteases that allow recognition of specific substrates are by far less understood. For presenilins, which attain their catalytic function in a high molecular weight γ-secretase complex, comprising either presenilin 1 or presenilin 2, nicastrin, presenilin enhancer 2 (PEN-2) and anterior pharynx-defective 1 (APH-1), it is suggested that a hydrophilic cavity within the large luminal/extracellular domain of nicastrin, the TM domains 2 and 6 and the PAL motif in TM 9 of presenilin, as well as a hybrid β-sheet formed between the substrate and presenilin close to the active site are crucial for substrate recognition[11,30–33]. In this context, the bulky luminal/extracellular domain of nicastrin is discussed to exclude substrates with large luminal/extracellular domains based on steric hinderance[34], explaining why presenilins favour substrates with luminal/extracellular domains (ectodomains) shorter than 50 amino acids[35]. Alpha-fold based structural predictions of the human SPPL proteases suggest that the spatial arrangement of the TM domains and the active site are comparable to that of presenilin[22]. However, in contrast to presenilin, the SPP/SPPL proteases do not require association with other proteins in high molecular weight complexes to gain proteolytic activity[2]. If intramembrane aspartyl proteases employ common mechanisms of substrate recognition, the role of nicastrin in the recognition of substrates by γ-secretase must be solved differently for SPP/SPPL proteases.

Similar to γ-secretase, SPPL2b also preferentially selects substrates with short ectodomains and many of its substrates require a preceding shedding by unrelated proteases, which significantly reduces the length of the substrate's ectodomain[36,37]. In contrast, SPPL3 accepts substrates with large and bulky ectodomains, and releases them from the membrane in a process termed non-canonical shedding[37,38]. SPP and SPPL2a generally prefer substrates with short ectodomains, but also accept the longer versions of certain substrates[37,39,40].

One major difference between the members of the SPP/SPPL protease family is their N-terminal domains[22]. While the SPPL2 subfamily members comprise a rather large and glycosylated N-terminal domain that exhibits homology with a protease-associated (PA) domain, SPPL3 essentially lacks an N-terminal domain[22]. The N-terminal domain of SPP is short and it is unknown whether it acquires a specific stable 3D structure[22]. PA domains are highly conserved in evolution from bacteria to humans and a role in regulating substrate access to proteases has been discussed[41,42]. Based on this, we put forward the hypothesis that the PA domains of the SPPL2 subfamily are involved in substrate selection and influence the differentiation between substrates with long and short ectodomains.

To address this experimentally, we chose tumour necrosis factor α (TNFα), a well characterized substrate of SPPL2a and SPPL2b[7,39], as model substrate. The processing of TNFα predominantly follows the principle of regulated intramembrane proteolysis (RIP)[7,8]. In this context, the TNFα homotrimeric ectodomain[43] is first released by sheddases of the disintegrin and metalloproteinase (ADAM) family resulting in secretion of soluble TNFα (sTNFα)[37,44]. The remaining N-terminal fragment (TNFα NTF) is membrane bound and the direct substrate for intramembrane cleavage by SPPL2a or SPPL2b[6,7]. Intramembrane cleavage starts with an initial cleavage at the luminal/extracellular membrane border releasing a small C-terminal TNFα peptide (C-peptide) and a long, still membrane bound ICD (TNFα ICD$_{long}$). To release the remaining membrane fragment (TNFα ICD$_{long}$), SPPL2a and SPPL2b apply multiple consecutive cleavages, also termed processivity, that can be monitored over time[6]. The trimmed TNFα ICD species are released to

the cytosol and are either rapidly degraded or can induce interleukin-12 secretion in immune cells[8,45]. In an alternative processing, TNFα is directly recognized by SPPL2a, resulting in secretion of a slightly larger soluble TNFα fragment (sTNFα(L2))[39]. The remaining membrane anchored TNFα (TNFα ICD$_{long}$) undergoes consecutive turnover by SPPL2a, as observed in context of RIP[39]. Since no TNFα ICD is observed upon co-expression of either SPP, SPPL3 or SPPL2c and TNFα, it is believed that these proteases do not cleave TNFα[7,8].

Using AlphaFold-predicted 3D structures[46,47], we performed a conservation analysis, which supported the hypothesis that the PA domains of the SPPL2 subfamily confer specific protein-protein interactions. Co-expression of chimeric proteases that comprise SPPL2a or SPPL2b catalytic domains and different SPPL N-terminal domains with TNFα revealed that the N-terminal PA domains of the SPPL2 proteases influence substrate recognition, in particular that of substrates with long bulky ectodomains, but are not the sole determinant.

## Results

### The surfaces of the SPPL2 PA domains show subtype specific conserved patches

To detect a potential canonical function of the PA domains of the SPPL2 subfamily members, we applied 3D analysis to determine the surface conservation between the PA domains of human SPPL2a, SPPL2b and SPPL2c (Fig. 1). To this end, we first aligned the linear protein sequences via Clustal Omega[48,49] and then colored the residues according to their conservation level in the 3-dimensional AlphaFold prediction[46,47]. Using this method, conserved patches on the surface that hint to specific protein interaction become apparent[50]. When we compared the three human subtypes SPPL2a, SPPL2b, and SPPL2c, however, we did not observe any highly conserved patches (Fig. 1a). Thus, most of the residues on the predicted surface are not conserved between the individual members of the human SPPL2 subfamily. On the other hand, when the surface of each subfamily member is analyzed for conservation across four mammalian species, highly conserved patches on the surface become apparent (Fig. 1b). Since human SPPL2a and SPPL2b share several substrates amongst them, but not with SPPL2c[22], we asked for conservation on the PA-domain of only SPPL2a and SPPL2b, which would explain their commonalities. We found a higher level of conservation, but not as high as in the cross-species comparison of the subtypes (Fig. 1c). This may hint to a role of the PA-domain in subtype-specific protein-protein interactions.

To experimentally address this, we designed based on AlphaFold predicted 3D structures of the SPPL-family members[22] six chimeric enzymes comprising the catalytic and C-terminal domain of either SPPL2a or SPPL2b combined with an N-terminal domain of the remaining three SPPL-family members (Fig. 2a, Supplementary (Suppl.) Fig. 1 & Suppl. Figure 2) and analysed their capacity to process TNFα. While all members of the SPPL2 subfamily comprise a rather large globular PA domain[22], the N-terminus of SPPL3 only includes 14 amino acids and thus, almost lacks an N-terminal domain. In contrast, SPP encompasses an N-terminal domain for which AlphaFold does not predict a distinct fold and therefore, we did not include SPP in this analysis. The domain border of the PA domains was determined according to the 3D-prediction and the corresponding residue position was applied to all variants, to ensure equivalent and proper folding.

### Expression levels of SPPL2a wt and SPPL2b wt are proportional to TNFα NTF reduction

Stable expression of wild-type (wt) SPPL2a and SPPL2b, as well as of the chimeric proteases, under a doxycycline inducible promotor in T-Rex® HEK293 cells lacking endogenous SPPL2a and SPPL2b expression (dKO cells[39]) revealed that all proteases are well expressed, exhibiting expected sizes and maturation, reflecting their published glycosylation status[1]. However, we observed differences in expression levels (Fig. 2b).

To evaluate whether and how differences in protease expression influence substrate turnover, we transiently co-expressed TNFα in

dKO cells comprising different stable expression levels of either SPPL2a wt (Suppl. Figure 3) or SPPL2b wt (Suppl. Figure 4) and monitored the TNFα cleavage products on Western Blot. Since TNFα NTF constitutes a direct substrate of SPPL2a and SPPL2b, its reduction relative to TNFα NTF in dKO cells reflects the efficiency of the initial SPPL2a or SPPL2b mediated cleavage (Suppl. Figures 3a & 4a). To also account for transfection efficiency of TNFα, the amount of TNFα NTF was normalized to the corresponding full length TNFα (TNFα FL). As expected, TNFα NTF levels in samples from cells with high SPPL2a expression were significantly lower than those in samples with low SPPL2a expression (Suppl. Figure 3a & b). Similarly, TNFα NTF levels were significantly different in cells with different SPPL2b expression levels (Suppl. Figure 4a & b). This confirms earlier data, which indicate that reduction of TNFα depends on the presence of catalytically active SPPL2a or SPPL2b[6–8]. To prove that the difference in TNFα NTF levels is proportional to the protease expression, the relative difference of protease expression was included in the calculation by multiplication with an expression factor (ExF). This resulted in similar TNFα NTF levels for cell clones with different SPPL2a (Suppl. Figure 3c) or SPPL2b (Suppl. Figure 4c) expression, respectively. This confirms that TNFα NTF reduction is proportional to the expression of the respective intramembrane protease and allows comparison of the catalytic efficiency of different SPPL-variants, resulting in the following correlation: The lower the TNFα NTF levels normalized to protease expression, the higher the efficiency of the initial TNFα cleavage.

TNFα ICD species result from the initial cleavage of TNFα NTF and are consecutively converted by the SPPL protease, resulting in the detection of multiple ICD species[6]. At a given time point, the total of all ICD species is proportional to the protease expression (Suppl. Figure 3d, e and Suppl. Figure 4d, e), and results in the following correlation: The higher the TNFα ICD levels normalized to protease expression, the higher the efficiency of the corresponding protease.

Furthermore, SPPL2a is also capable of directly cleaving TNFα FL acting as non-canonical sheddase[37,39]. Thus, the resulting soluble TNFα (sTNFα L2) is a direct product of SPPL2a cleavage that is secreted to the conditioned media. In support of this, the amount of secreted sTNFα L2 is proportional to the SPPL2a expression level (Suppl. Figure 3f, g). This demonstrates that reduction of TNFα NTF and production of TNFα ICDs, as well as of sTNFα L2, is proportional to the protease expression and allows the use of these TNFα species normalized to transfection efficiency and protease expression levels to compare the catalytic efficiency of SPPL2a and SPPL2b variants with different expression levels.

### The SPPL N-terminal domains differentially affect TNFα processing by SPPL2a and SPPL2b in cells

Efficient proteolytic processing of a substrate in intact cells requires not only compatibility of substrate and enzyme, but also co-localization of substrate and enzyme in the same subcellular compartment. Thus, we investigated whether the stably expressed chimeric proteases exhibit the same localization as their corresponding wt protease. Using confocal microscopy and immunofluorescence labeling via the C-terminal HA-tag of the wt and chimeric proteases, we determined their subcellular localization (Fig. 3). Confirming earlier findings[16], SPPL2a wt predominantly localized to endosomal/lysosomal compartments (Fig. 3a) and SPPL2b wt was detected mainly on the plasma membrane (Fig. 3b). While all SPPL2a chimeras depicted a similar subcellular distribution to SPPL2a wt (Fig. 3a), only the SPPL2b chimeric protease comprising the SPPL2a PA domain (2a/2b) was still sorted to the plasma membrane (Fig. 3b). Presence of either the PA domain of the ER-localized SPPL2c (2c/2b) or the short N-terminus of SPPL3 (3/2b) resulted in predominantly ER localization of the respective SPPL2b chimeric protease (Fig. 3b–d). Those chimeric proteases that

**Fig. 1 | Conservation analysis of the SPPL2 PA domains. a** Sequences of human SPPL2a, SPPL2b, and SPPL2c were aligned with Clustal Omega[48,49] and conservation mapped onto the AlphaFold[46,47] predicted surface of the N-terminal PA domain. The globular PA domains are depicted as solvent accessible surfaces. Side chains that are identical are marked in dark green, chemically closely related side chains in pale green, and loosely related side chains in yellow. The helices and loops of the catalytic domains are drawn in grey. The C-terminal domains are omitted. No highly conserved surface patches are observed in the cross-subtype analysis. **b** SPPL2a, SPPL2b and SPPL2c from four different mammalian species (pig, mouse, Tasmanian devil, human) were aligned and mapped onto the structures with the same color code as in (**a**) to identify conserved subtype specific surface patches. The conservation of SPPL2s was plotted onto the structure prediction of the respective human subtype in the cross-species comparison. Structure representations were drawn in Pymol[64]. The surface residues of the individual SPPL2 proteases are highly conserved among mammalian species. **c** Sequences of human SPPL2a and SPPL2b were aligned and mapped onto the structures with the same color code as in (**a**). Human SPPL2a and SPPL2b share conserved surface residues.

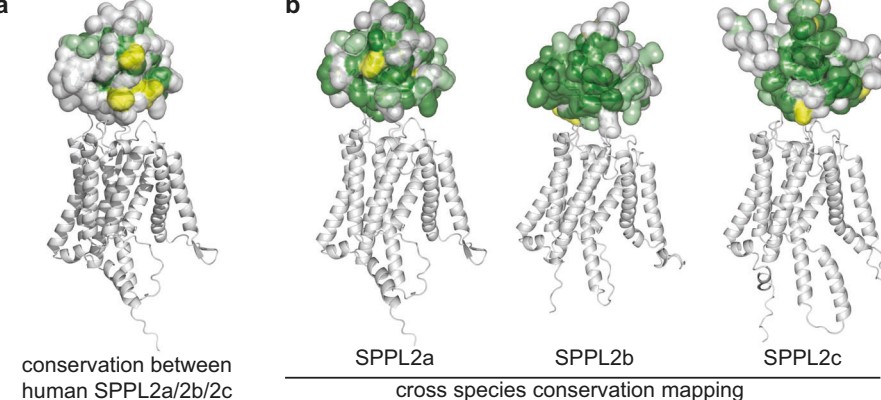

conservation between human SPPL2a/2b/2c

SPPL2a          SPPL2b          SPPL2c

cross species conservation mapping

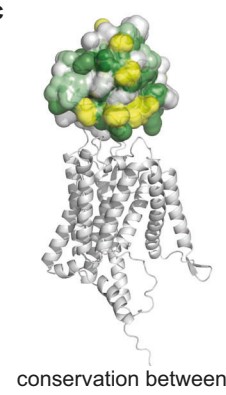

conservation between human SPPL2a/2b

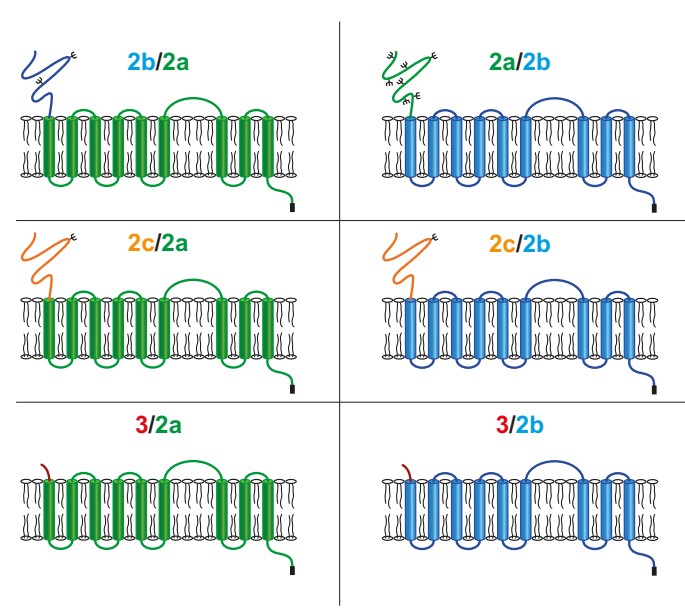

**Fig. 2 | Chimeric SPPL proteases. a** *Schematic representation of SPPL family members and chimeric proteases*. Protein domains of SPPL2a are depicted in green, those of SPPL2b in blue, SPPL2c protein domains in orange and of SPPL3 in red. Predicted N-glycosylation sites are marked with black tree-like structures and the HA-tag as black boxes. **b** *Expression of SPPL family members and chimeric proteases*. SPPL2a wt (2a wt), SPPL2b wt (2b wt) and all chimeric proteases were stably expressed under a doxycycline inducible promotor in dKO cells. Proteins were visualized using the anti-HA antibody 3F10. Calnexin was used as loading control.

localized like their respective wt, were analyzed for their capacity of TNFα NTF processing. To this end, we transiently co-expressed TNFα FL with either the SPPL2 wt proteases or the selected chimeric enzymes and monitored TNFα NTF and TNFα ICD on Western Blot (Fig. 4a). Considering the variation in protease expression, quantification of TNFα NTF revealed that replacing the SPPL2a N-Terminal domain by that of SPPL2c (2c/2a) or SPPL3 (3/2a) significantly reduced the efficiency of TNFα NTF cleavage, while the N-terminus of SPPL2b (2b/2a) did not significantly affect the efficiency of the SPPL2a mediated cleavage (Fig. 4b). In contrast, replacement of the SPPL2b N-terminal domain by that of SPPL2a (2a/2b) significantly increased the efficiency of SPPL2b mediated TNFα NTF cleavage (Fig. 4c). As expected, the SPPL2a chimeric protease comprising the SPPL2c PA domain (2c/2a) also depicted reduced TNFα ICD production relative to the respective wt proteases, while the SPPL2b PA domain (2b/2a) did not affect TNFα ICD production of SPPL2a (Fig. 4d). Contrary to our expectations, the SPPL3 N-terminal domain (3/2a) did not reduce TNFα ICD production of SPPL2a (Fig. 4d), although TNFα NTF was significantly enriched in these cells when protease expression was considered (Fig. 4b).

This may point to a change in non-canonical shedding versus RIP by SPPL3/2a compared to SPPL2a wt and will be addressed in more detail below. Fusion of the SPPL2a PA domain to SPPL2b (2a/2b) did not reveal a significant increase in TNFα ICD production (Fig. 4e), as would have been expected from the reduction in TNFα NTF levels (Fig. 4c). This may either be due to fast subsequent turnover of TNFα ICD species compared to the TNFα NTF and, thus, a more inaccurate quantification, or indicate a change in processivity of the chimeric protease, which will be addressed in more detail below.

### The SPPL N-terminal domains differentially affect TNFα processing by SPPL2b in vitro

To address the capability of TNFα cleavage by those SPPL2b chimeras that depict a change in localization in intact cells, we utilized our recently established in vitro assay that monitors initial cleavage of TNFα NTF independently of protease localization[51].

SPPL2b wt or the SPPL2b chimeras (Fig. 2a & Suppl. Figure 2) were transiently expressed in dKO cells and TNFα FL was expressed in separate

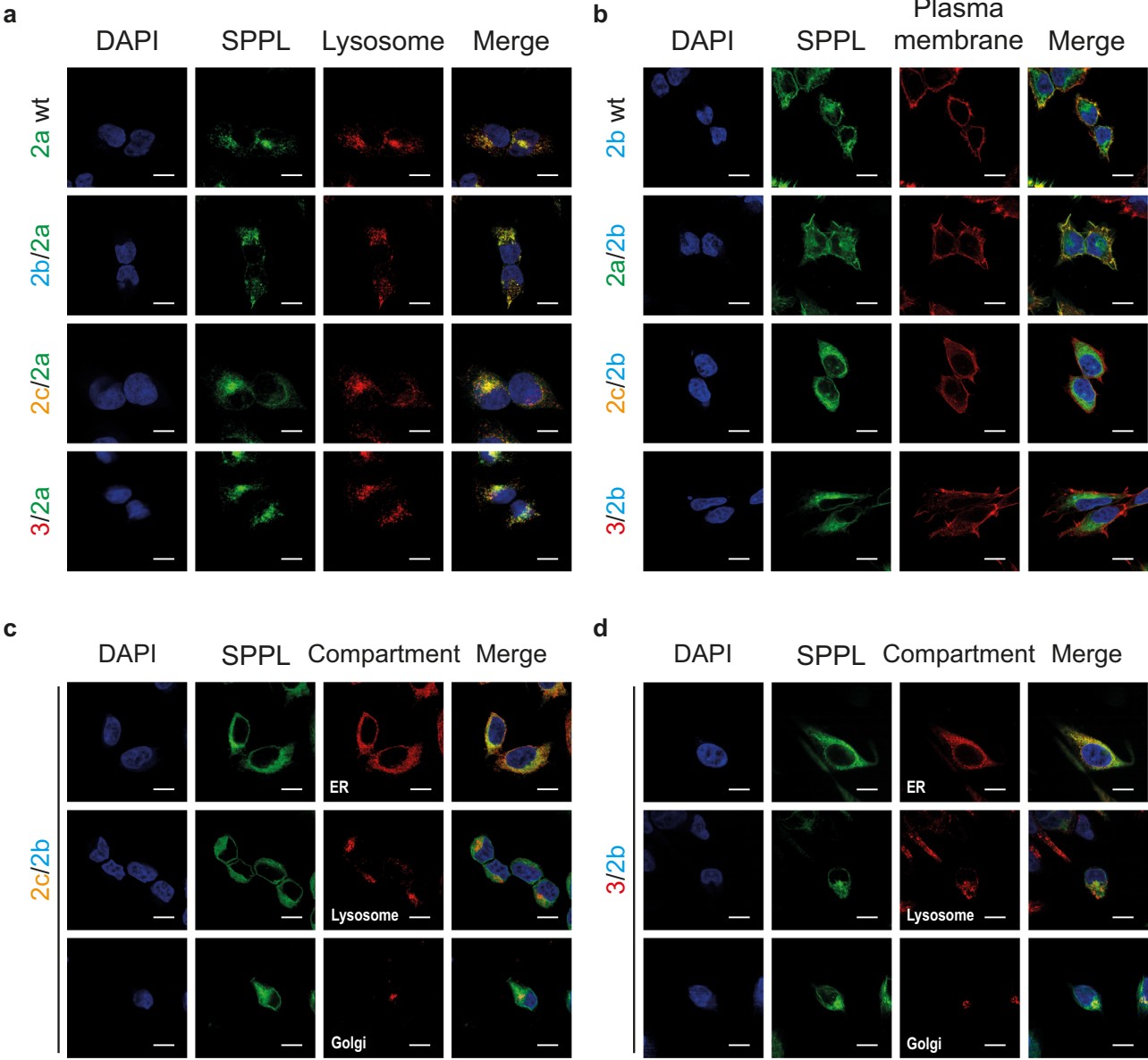

**Fig. 3 | Subcellular localization of chimeric proteases.** Wt and chimeric proteases were visualized using 3F10-fluorescein targeting the C-terminal HA-tag. For labeling of lysosomes Lamp1 (D2D11) was used, actin staining (phalloidin) served as plasma membrane marker, BiP for visualization of the ER and DAPI for nuclear staining. Nuclei are depicted in blue, SPPL wt and chimeric proteases in green and cells. Proteases and substrate were detergent solubilized and incubated at 37 °C for 2 h. Protease expression and substrate turnover were monitored on Western Blot (Fig. 5a). Corroborating our finding in intact cells, the chimeric protease comprising the SPPL2a PA domain and the catalytic part of SPPL2b (2a/2b), depicted a significantly stronger reduction of TNFα NTF compared to SPPL2b wt, after normalization to protease amount (Fig. 5b). Presence of the SPPL2c PA domain (2c/2b) significantly reduced SPPL2b mediated TNFα NTF turnover (Fig. 5b), similar to the effect of the SPPL2c/2a chimera observed in intact cells (Fig. 4b). Interestingly, the SPPL3 short N-terminus (3/2b) also enhanced TNFα NTF turnover (Fig. 5b). This distinguishes SPPL2b from SPPL2a, which exhibited reduced TNFα NTF cleavage in the presence of the SPPL3 N-terminus (Fig. 4b; 3/2a).

Together, these data indicate that the SPPL2a PA domain increases the efficiency of TNFα NTF processing compared to the presence of the SPPL2b PA domain. The presence of the SPPL2c PA domain significantly reduced activity of both, SPPL2a and SPPL2b, on TNFα NTF and, thus, seems to be

cell compartments in red. White bar = 10 μm. **a** SPPL2a wt and all SPPL2a chimeric proteases colocalized with the lysosomal marker. **b** SPPL2b wt and the SPPL2a/2b chimera colocalized with the plasma membrane marker, while the other chimera are not observed at the plasma membrane. **c, d** SPPL2c/2b and SPPL3/2b mainly colocalize with the ER marker.

the least favorable domain for both proteases. Interestingly, the short N-terminus of SPPL3, which is so far the only known bona fide non-canonical sheddase of the SPP/SPPL family[37], affected TNFα processing by SPPL2a and SPPL2b differently. While SPPL2a NTF processing by SPPL2a was hampered, that of SPPL2b was slightly increased. However, production of TNFα ICDs by SPPL2a was not affected.

### The SPPL N-terminal domains affect the kinetics of TNFα NTF processing

Having demonstrated that the SPPL N-terminal domains affect the processing of TNFα in vitro and in intact cells, we next asked whether they also influence the kinetics of TNFα NTF turnover. To this end, we used a membrane-based cleavage assay published earlier[6], which allows monitoring the time dependent conversion of TNFα NTF independently of the subcellular localization of protease and substrate. Membranes from cells co-expressing full length TNFα and wt or chimeric proteases were isolated,

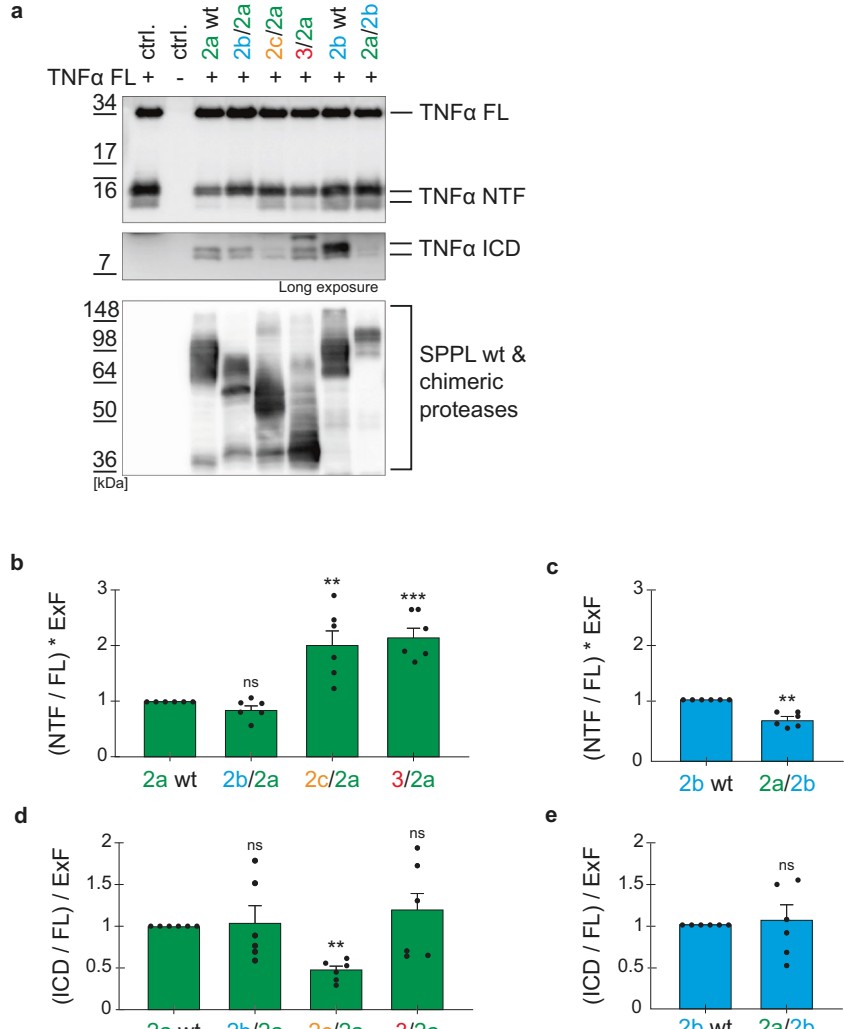

**Fig. 4 | TNFα cleavage altered by N-terminal SPPL chimeras. a** DKO cells (ctrl.+) and dKO cells overexpressing the indicated wt or chimeric proteases were transiently transfected with full length TNFα (TNFα FL). Non-TNFα transfected cells. (ctrl -) were included. Membrane bound TNFα species were analyzed on Western Blot using the anti-FlagM2 antibody. An antibody against the HA-tag (3F10) was used to detect SPPL expression. **b, c** Densitometric quantification of non-cleaved TNFα NTF, as depicted in (**a**). Normalization to TNFα FL eliminated transfection variations. Expression differences between chimeric and wt proteases are eliminated by multiplication with the expression factor (ExF). Resulting values were all normalized to their corresponding wt protease sample. (**d&e**) Densitometric quantification of total TNFα ICD, as depicted in (**a**). Normalization as in (**b, c**). Expression differences between chimeric and wt proteases are eliminated by division with the ExF. **b–e** Mean + SEM, unpaired, two-tailed one sample t-tests of log-transformed ($\log_2$) values. ns not significant, **$p < 0.01$, ***$p < 0.001$ (**b**: p of 2b/2a = 0.1191, p of 2c/2a = 0.0041, p of 3/2a = 0.0002; **c**: p of 2a/2b = 0.0031; **d**: p of 2b/2a = 0.8154, p of 2c/2a = 0.0011, p of 3/2a = 0.8419; **e**: p of 2a/2b = 0.8872), $n = 6$ independent experiments. The ExF is the mean ($n = 6$) of the ratio between the expression of the individual chimera and its corresponding wt protease.

homogenized, incubated at 37 °C and reduction of TNFα NTF over time was monitored on Western Blot. Additionally, the production and conversion of TNFα ICD species was visualized (Suppl. Figure 5a). To quantify the reduction of TNFα NTF over time, the amount of TNFα NTF at every individual time point was normalized to calnexin as a loading control and depicted relative to the respective incubation time (Suppl. Figure 5b). Linear fit resulted in a slope that reflected the average speed of TNFα NTF reduction over time. The faster the turnover, the larger the absolute value of the slope (Suppl. Figure 5b). To test whether protease expression also quantitatively correlates with the speed of TNFα NTF turnover, single cell clones with different expression levels of SPPL2a wt were compared (Suppl. Figure 5). As expected, TNFα NTF reduction was significantly faster in cell clones with high protease expression (Suppl. Figure 5a–c) and the speed was directly proportional to the expression factor (ExF) (Suppl. Figure 5d). This confirms that TNFα NTF turnover quantitatively correlates with the amount of protease present in the cell. The turnover is reflected in the

absolute slope: the lower it is, the less efficient is the time-dependent turnover of TNFα NTF. Based on this, we monitored and quantified TNFα NTF turnover in membranes co-expressing TNFα FL and the chimeric proteases compared to the respective wt proteases (Figs. 6 & 7). Time dependent turnover of TNFα NTF by SPPL2a was significantly reduced in the presence of either the SPPL2b (2b/2a) or the SPPL2c (2c/2a) PA domain, while the SPPL3 N-terminus had no significant impact on the speed of TNFα NTF turnover. (Fig. 6d). This suggests that kinetics of the initial TNFα cleavage by SPPL2a is most efficient in presence of the wt PA domain or the SPPL3 short N-terminus. Interestingly, while the reduction in TNFα NTF turnover speed by the SPPL2b/2a chimeric protease seems to be compensated when protease and substrates are co-expressed in intact cells (Fig. 4b), where time is not a limitation for cleavage, the presence of the SPPL2c PA domain impairs the initial TNFα NTF cleavage in both scenarios (Figs. 4b & 7d). The short N-terminus of SPPL3, which reduced cleavage of TNFα NTF by SPPL2a in the cell (Fig. 4b; 3/2a), did not affect the TNFα NTF turnover in cellular

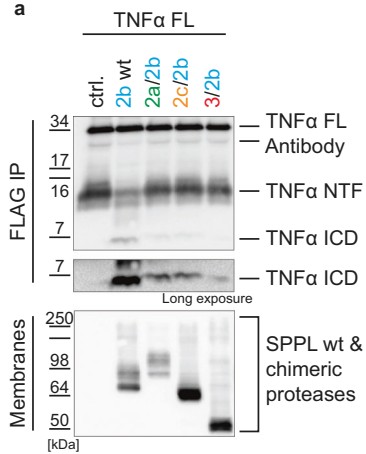

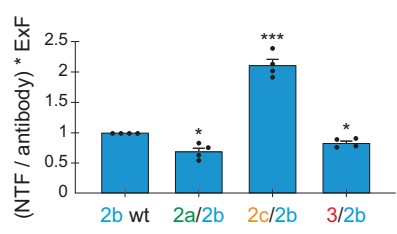

**Fig. 5 | In vitro cleavage of TNFα NTF generated from TNFα FL by SPPL2b chimeric proteases. (a)** DKO cells were either non-transfected (ctrl.) or transiently transfected with SPPL2b wt or the indicated chimeric proteases. In vitro cleavage of separately expressed full length TNFα (TNFα FL) was carried out and TNFα species were immunoprecipitated (FlagM2) and analyzed on Western Blot (polyclonal anti-Flag antibody). SPPL proteases were detected in solubilized membranes with an antibody against the HA-tag (3F10). **(b)** Densitometric quantification of non-cleaved TNFα NTF, as depicted in (a). Normalization to the antibody signal eliminates experimental variations as described earlier[51]. Expression differences between chimeric and wt proteases are accounted for by multiplication with ExF. Resulting values were all normalized to the SPPL2b wt protease sample. Mean + SEM, unpaired, two-tailed one sample t-tests of log-transformed ($\log_2$) values. ns=not significant, $*p < 0.05$, $***p < 0.001$ (p of 2a/2b = 0.026, p of 2c/2b = 0.0006, p of 3/2b = 0.0242), $n = 4$ independent experiments. The ExF is the mean ($n = 4$) of the ratio between the expression of the individual chimera and SPPL2b wt.

membranes over time (Fig. 6c, d), this is in line with the unaffected generation of TNFα ICD species in the cell (Fig. 4d). In addition, the appearance of shorter ICD species tends to occur later in presence of all chimeric SPPL2a proteases compared to SPPL2a wt (Fig. 6a–c). This may indicate that also consecutive processing of the substrate directly or indirectly depends on the enzymes N-terminal domain.

The SPPL2c PA domain also significantly reduced the speed of SPPL2b (2c/2b) mediated TNFα NTF turnover, while presence of the SPPL2a (2a/2b) PA domain or SPPL3 (3/2b) N-terminus did not affect the kinetics of TNFα NTF turnover (Fig. 6d). Interestingly, generation of TNFα ICD species in the presence of the SPPL2a/2b chimeric protease (Fig. 7a) is comparable to that of SPPL2b wt, while the other chimeric enzymes elicited later occurrence of TNFα ICD species (Fig. 7b, c), similar to what we observed for all SPPL2a chimeras (Fig. 6). This suggests, that SPPL2b-mediated consecutive TNFα cleavage is most efficient in presence of either the wt PA domain or of the PA domain of SPPL2a.

### The SPPL N-terminal domains do not alter the SPPL2a/b cleavage sites in TNFα

To ensure that chimeric SPPL2 proteases do not alter cleavage sites within the TNFα TM domain, we carried out MALDI-TOF mass spectrometry of the TNFα ICD species (Fig. 8 and Suppl. Figure 6). The known major SPPL2a and SPPL2b cleavage products of TNFα[7,39] are detected regardless of whether a wt or a chimeric protease is expressed (Fig. 8), indicating that none of the chimeric proteases affects the precision of TNFα cleavage. However, the ratio of the longer TNFα ICD species (S39) relative to the shorter species (R28) is increased in all SPPL2a chimeras at the time of measurement, corroborating a slower conversion of the TNFα ICD species compared to the wt protease (Fig. 8). Reduced processivity is also detected in presence of the SPPL2b chimeras comprising the SPPL2c PA domain (2c/2b) or SPPL3 N-terminus (3/2b), but is less pronounced in presence of the SPPL2a PA domain (2a/2b) (Fig. 8). This is in line with the analysis of the time dependent TNFα turnover, which depicts delayed TNFα ICD generation for all chimeric proteases, except for the 2a/2b chimeric protease (Figs. 6, 7).

Taken together, the kinetics of initial and processive cleavage by SPPL2a is slowed by the presence of the SPPL2b PA domain (2b/2a). However, while this effect is compensated upon permanent co-expression with the substrate, the effect of the SPPL2c PA domain (2c/2a) is not

(Fig. 4b, d). This suggests that the cleavage impairment induced by the SPPL2c PA domain is stronger than that of the SPPL2b PA domain. Similarly, the SPPL2c PA domain also significantly reduced the speed of TNFα NTF processing by SPPL2b (Fig. 7b, d), demonstrating that the SPPL2c PA domain is the least efficient for processing of TNFα.

In contrast, the kinetics of initial TNFα NTF cleavage by SPPL2b was not significantly affected by the presence of the SPPL2a PA domain (2a/2b) (Fig. 7a, d), although the initial TNFα NTF cleavage in the cell was more efficient (Fig. 4a, c). Interestingly, the total amount of TNFα ICD levels generated by SPPL2a/2b (Fig. 4e), as well as the conversion of the TNFα ICDs over time (Fig. 7a), were comparable to SPPL2b wt. This indicates that while the SPPL2a PA domain enhances the initial cleavage of TNFα NTF by SPPL2b, when time is not a limiting factor, the processive conversion of TNFα ICD appears to depend solely on the TNFα ICD presence in the protease's active center, which is continuously further processed.

### SPPL3 processes TNFα as non-canonical sheddase

Since the short N-terminus of SPPL3 (3/2a) significantly reduced SPPL2a mediated TNFα NTF turnover in cells (Fig. 4b) but affected neither the TNFα ICD production (Fig. 4d) nor the kinetics of the initial cleavage or the processivity (Fig. 6c, d), we asked whether this is related to a shift in ratio between RIP and non-canonical shedding. To address this, we first tested whether SPPL3 wt is capable of recognizing and cleaving TNFα FL, since previous work showed that SPPL3 wt does not produce TNFα ICD species when co-expressed with TNFα FL[7,8]. To exclude the non-canonical shedding activity of SPPL2a[39] in this analysis, we expressed SPPL3 in dKO cells, which lack endogenous SPPL2a/2b expression, and analyzed TNFα cleavage products on Western Blot (Fig. 9a). Corroborating our earlier findings[7,8], we did not detect any TNFα ICD species in these cells (Fig. 9a), and TNFα NTF levels did not change upon expression of SPPL3 (Fig. 9b). However, when SPPL3 was overexpressed, we detected substantial amounts of a larger TNFα soluble fragment (sTNFα L3) at a similar molecular weight as the product of non-canonical SPPL2a-shedding (Fig. 9a, c). This effect was not observed with SPPL2b overexpression, which, even when overexpressed, shows only minor non-canonical shedding activity[39]. These data suggest, that SPPL3 can act as a non-canonical sheddase at the luminal membrane-border of TNFα FL, resulting in a soluble sTNFα L3 fragment, which is slightly larger than the ADAM produced sTNFα, and a longer membrane bound TNFα ICD, which cannot be reliably distinguished from the ADAM-generated

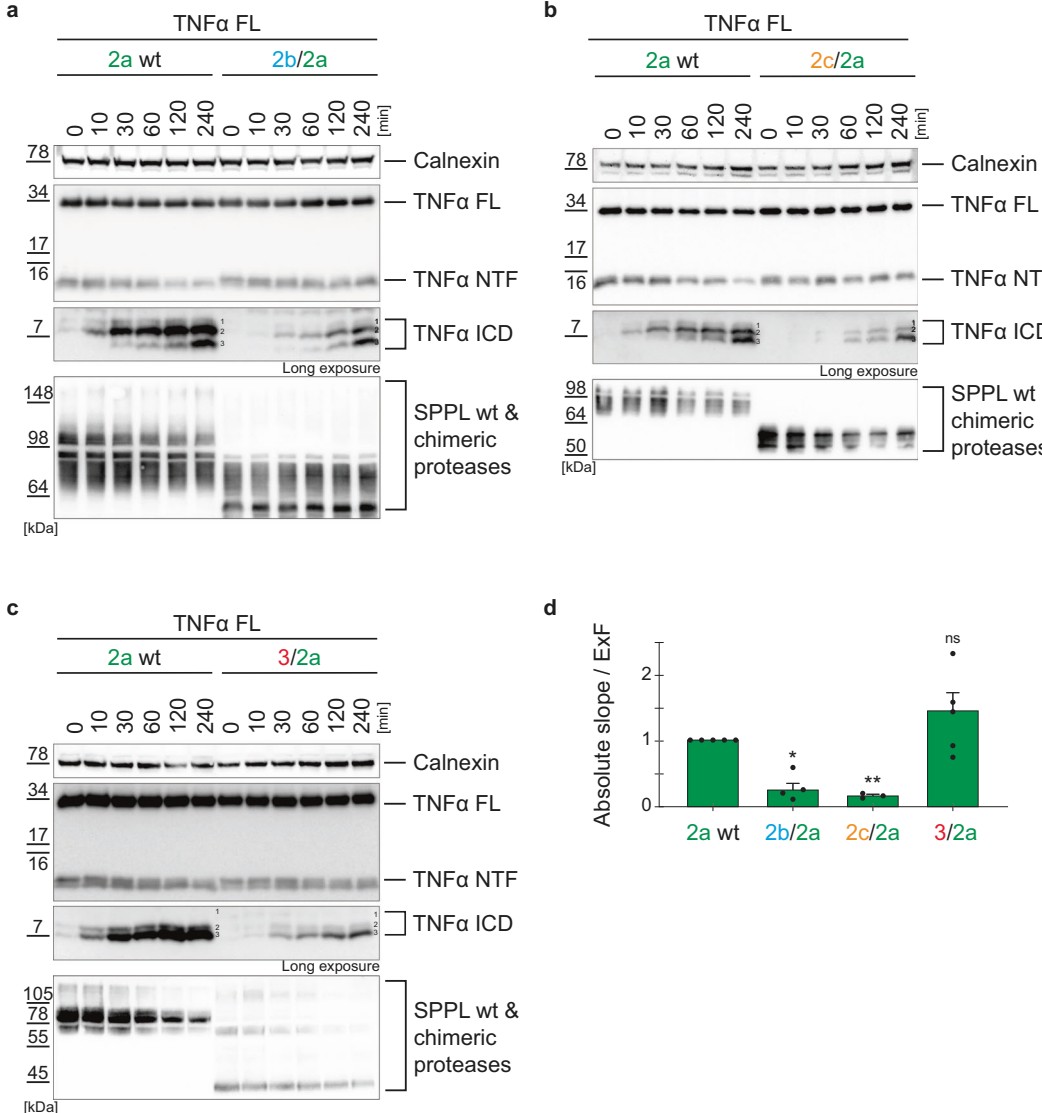

**Fig. 6 | Kinetics of TNFα turnover by SPPL2a chimeras. a–c** DKO cells (ctrl.) and dKO cells stably overexpressing either SPPL2a wt or the indicated chimeras were transiently transfected with full length TNFα (TNFα FL). Membrane isolates were incubated shaking at 37 °C for the indicated time periods and TNFα species were analyzed on Western Blot using FlagM2 antibody. Calnexin served as loading control. **d** The TNFα NTF-reduction over time is represented as absolute slope/ExF (calculation shown in Suppl. Figure 5b). Expression differences between chimeric and wt proteases are accounted for by division with the ExF at time point 0 min. Resulting values were all normalized to the SPPL2a wt protease sample. Mean + SEM, unpaired, two-tailed one sample t-tests of log-transformed ($\log_2$) values. ns=not significant, \*$p < 0.05$, \*\*$p < 0.01$ (p of 2b/2a = 0.0262, p of 2c/2a = 0.0048, p of 3/2a = 0.2336), $n = 3$–5 independent experiments (**a**: $n = 4$, **b**: $n = 3$, **c**: $n = 5$). The ExF is the mean (**a**: $n = 4$, **b**: $n = 3$, **c**: $n = 5$) of the ratio between the expression of the individual chimera and SPPL2a wt.

TNFα NTF and therefore accounts for the stable TNFα NTF levels (Fig. 9b). In contrast to SPPL2a and SPPL2b, SPPL3 seems not to be capable of further processing the membrane bound TNFα ICD, since increased expression of SPPL3 does not lead to the detection of smaller ICDs (Fig. 9a, b).

### The N-terminal domains of non-canonical sheddases reduce the capacity of SPPL2a and SPPL2b to recognize ADAM-generated TNFα NTF

Based on this, we analyzed non-canonical shedding activity in the cell lines co-expressing TNFα FL and those chimeras that localize to the same compartment as the corresponding wt protease (Fig. 9d). Quantification of the non-canonical shedding products (sTNFα L2‖3) normalized to the expression level of the respective protease revealed no change in secretion of sTNFα L2‖3 for all the SPPL2a chimeric proteases (Fig. 9e), but an increase in non-canonical shedding of the SPPL2a/2b chimera relative to SPPL2b wt (Fig. 9f). Together, this suggests that the SPPL3/2a chimeric protease has a

similar non-canonical shedding activity as SPPL2a wt. However, in contrast to SPPL3 wt, it can consecutively process both TNFα NTF and TNFα ICD$_{long}$, the direct membrane-bound product of non-canonical shedding. This might explain, why we do not detect changes in total amount of TNFα ICD species upon expression of the SPPL3/2a chimera (Fig. 4d). In contrast to SPPL2a wt, the SPPL3/2a chimera apparently recognized the ADAM-generated TNFα NTF less efficiently, explaining the increase of this species relative to SPPL2a wt in intact cells (Fig. 4b).

This raises the question, whether the SPPL2b chimeric protease with the short SPPL3 N-terminus (3/2b) shows a similar behavior on ADAM-generated TNFα NTF. However, since the chimeric SPPL3/2b no longer localizes to the plasma membrane, we are not able to address this in the cellular system by directly detecting the non-canonical shedding products. Thus, we utilized our SPPL2b in vitro assay and instead of TNFα FL provided TNFα NTF as a substrate, mimicking the ADAM cleaved TNFα (Fig. 10). As expected, the chimeric SPPL2b protease that contains the

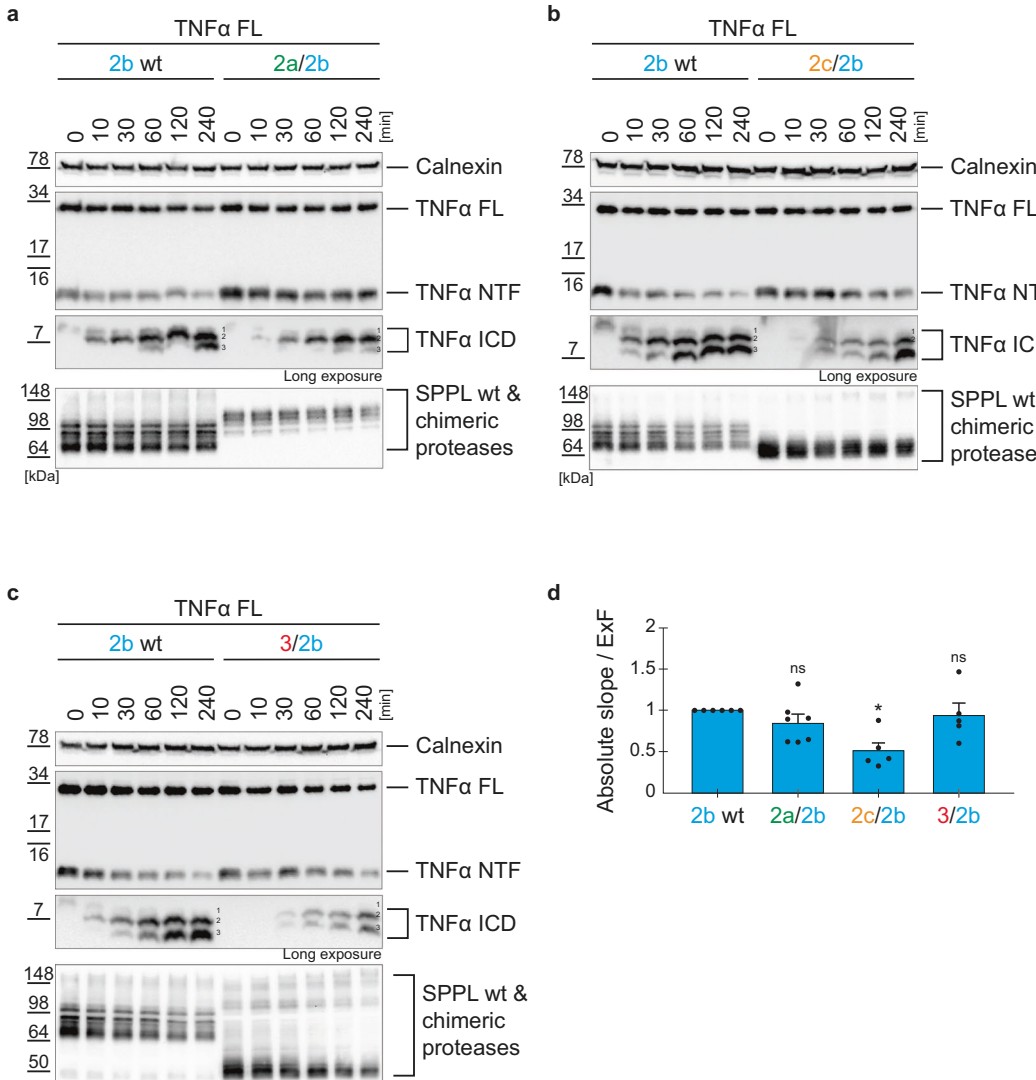

**Fig. 7 | Kinetics of TNFα turnover by SPPL2b chimeras. a–c** DKO cells stably overexpressing either SPPL2b wt or the indicated chimeras were transiently transfected with full length TNFα (TNFα FL). Membrane isolates were incubated shaking at 37 °C for the indicated time periods and TNFα species were analyzed on Western Blot using FlagM2 antibody. Calnexin served as loading control. **d** The TNFα NTF-reduction over time is represented as absolute slope/ExF (calculation shown in Suppl. Figure 5b). Expression differences between chimeric and wt proteases are accounted for by division with the ExF at time point 0 min. Resulting values were all normalized to the 2b wt protease sample. Mean + SEM, unpaired, two-tailed one sample t-tests of log-transformed ($\log_2$) values. ns not significant, *$p < 0.05$ (p of 2a/2b = 0.1265, p of 2c/2b = 0.0131, p of 3/2b = 0.5313), $n = 4$–7 independent experiments (**a**: $n = 7$, **b**: $n = 4$, **c**: $n = 4$), $n = 4$–7 (**a**: $n = 7$, b: $n = 4$, c: $n = 4$). The ExF is the mean (**a**: $n = 7$, **b**: $n = 4$, **c**: $n = 4$) of the ratio between the expression of the individual chimera and SPPL2b wt.

SPPL2c PA domain (2c/2b), cleaved TNFα NTF significantly less efficiently (Fig. 10b), further supporting our finding that the SPPL2c PA domain impairs TNFα processing strongest of all N-terminal domains tested.

Instead, the combination of SPPL2a (2a/2b) and SPPL3 (3/2b) N-termini with SPPL2b processed membrane bound species derived from TNFα FL more efficiently (Fig. 5b) than SPPL2b wt, while the cleavage of the solely ADAM derived TNFα NTF is reduced (Fig. 10b). This indicates that these chimeras might discriminate between TNFα NTF and TNFα ICD$_{long}$. The increased non-canonical shedding by the SPPL2a/2b chimeric proteases (Fig. 9f) is likely also present in the in vitro assay comprising TNFα FL as substrate. However, the detection of soluble TNFα species is not feasible in the assay setup. Together, our data suggest that SPPL2b chimera with either the SPPL2a PA domain or the short N-terminus of SPPL3 prefer membrane bound TNFα species that result from non-canonical sheddases as substrates, over those resulting from ADAM cleavage. Interestingly, the combination of the SPPL2a PA domain with SPPL2b linked the capability of SPPL2a non-canonical shedding with the strong processivity of SPPL2b, but

reduced recognition of ADAM-generated TNFα NTF the least (Figs. 4c, e, 5b, 7d, 10b). This explains why the presence of the SPPL2a PA domain overall increased turnover of TNFα FL by SPPL2b in all setups tested.

## Discussion
In this study we tested the hypothesis whether the N-terminal domains of the SPPL-family members, in particular the PA domains of the SPPL2 subfamily, are involved in recognition and processing of TNFα. Indeed, our results indicate that the SPPL N-terminal domains are involved in substrate recognition and even in the discrimination of substrates that only slightly differ in their luminal/extracellular domain. None of the SPPL N-terminal domain exchanges completely abolished processing of TNFα, indicating that substrate recognition by SPPL proteases is also substantially determined by other domains of the enzymes. For γ-secretase, it was suggested that a hybrid β-sheet formed between a β-strand in the substrate's cytoplasmic domain and another β-strand of the enzyme close to its active site are crucial for substrate binding[30,31]. Since presenilins and their

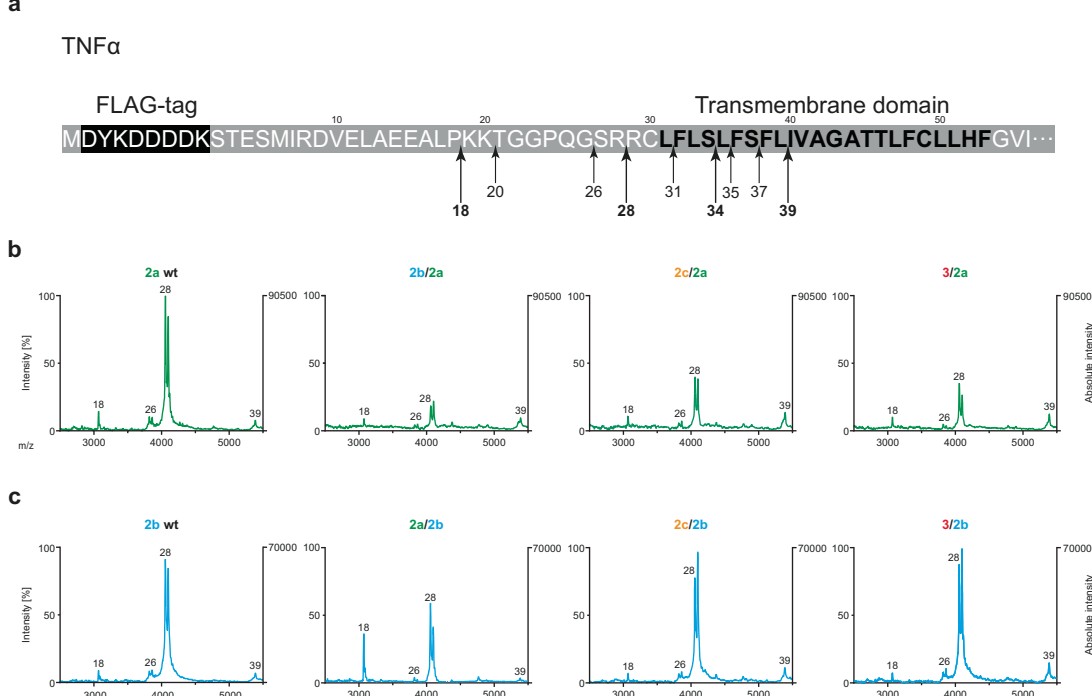

**Fig. 8 | Cleavage sites of chimeric proteases. a** Amino acid sequence (single letter code) and schematic representation of the TNFα N-terminus. Arrows indicate the known SPPL2 cleavage sites[39]. **b, c** Mass spectrometric analysis of TNFα ICDs generated from either SPPL2a (**b**) or SPPL2b chimeric proteases (**c**) compared to the respective wt protease. Membranes from dKO cells co-expressing wt or chimeric substrates both display inverted topologies compared to SPPL proteases[22], SPPL proteases and TNFα FL were isolated and incubated for 30 min at 37 °C. TNFα species were isolated and analyzed by MALDI-TOF. Numbers indicate the position of the most C-terminal amino acid of the respective cleavage product as indicated in (**a**). Control spectra are shown in Suppl. Figure 6.

the relative positioning of the substrate's TM helix within the active enzyme is likely the same[22]. Thus, it is tempting to speculate that formation of a similar hybrid β-sheet between TNFα and SPPL2a or SPPL2b is one critical determinant for successful substrate recognition, independent of the N-terminal enzyme domain. However, as a final proof for the relevance of the hybrid β-sheet in substrate recognition by SPPL proteases, complex structure determination of substrate and enzyme by cryo-electron microscopy or crystallization would be required.

Experimentally determined γ-secretase structures with peptides of APP and Notch, that comprise their TM domains and short parts of their juxtamembrane domains, also indicate that the substrate interacts with a hydrophilic cavity of nicastrin via a short helix or loop[30,31]. According to current knowledge, SPPL proteases do not form hetero-complexes with nicastrin-like proteins, therefore the PA domains of SPPL2a and SPPL2b might take over this function. PA domains are evolutionary highly conserved and are thought to regulate substrate access in proteases as one potential function[41,42]. Presence of the SPPL2c PA domain impaired TNFα processing by both SPPL2a and SPPL2b regarding initial cleavage, kinetics and processivity in cells and in vitro (Fig. 4b, d, Fig. 5b, Fig. 6b, d, Fig. 7b, d, Fig. 9e & Fig.10b), supporting a function of the PA domain in context of substrate recognition.

Interactions of proteins are modulated by pH and salt concentration of the surrounding milieu[52,53]. Like nicastrin, the PA domains of the SPPL proteases face the lumen of the compartment they are localizing to. The PA domain of SPPL2a physiologically faces an acidic lysosomal pH, while that of SPPL2c is exposed to a rather neutral pH of the ER. Thus, impaired function of the SPPL2c PA domain in context of the SPPL2a chimera might be explained by the pH difference caused by lysosomal localization of the chimeric protease. However, the pH in the in vitro experiments is close to the physiological pH in the ER, and still the SPPL2c PA domain significantly reduced TNFα NTF processing, indicating that differences in pH are not the

sole explanation for reduced substrate interaction and likely additional, so far enigmatic, differences in the 3D-structure of the PA domains are causal.

Since co-expression of SPPL3 and TNFα did neither result in TNFα NTF reduction nor in TNFα ICD production, TNFα was not considered an SPPL3 substrate[7,8]. A few years later, SPPL3 has been described as a bona fide non-canonical sheddase[14,15,37] and as such we now provide evidence that it can cleave full-length TNFα, resulting in the secretion of a soluble C-terminal TNFα fragment (sTNFα L3), which is slightly larger than the product of ADAM cleavage (Fig. 9a, c). The cleavage sites of SPPL3 in other known substrates, like GnT-V or other glycan modifying enzymes, map to the membrane border at the C-terminal end of the substrate's TM domain or even to the luminal juxtamembrane border[14,15,54]. Assuming this holds true for the processing of TNFα by SPPL3, the corresponding TNFα fragment is still membrane anchored and differs by only about 15 to 20 amino acids from the TNFα N-terminal fragments generated by ADAM-protease cleavage[55-57]. Due to this small difference and based on the fact that also ADAM-proteases generate multiple TNFα NTF species, it is not possible to separate these TNFα species on SDS-gels. Therefore, and consistent with our earlier data, we did not detect reduction of TNFα NTF (Fig. 9b). Also, no production of TNFα ICD was detected, corroborating that SPPL3 is not capable of consecutively processing the membrane bound TNFα species. Interestingly, the chimeric proteases comprising the SPPL3 N-terminal domain fused to either SPPL2a (3/2a) or SPPL2b (3/2b), produced TNFα ICD species and depicted processive cleavage at a kinetic that was not significantly different from that of the respective wt protease (Figs. 4a, d, 6 & 7). This provides evidence that the ability of consecutively cleaving transmembrane domains is mainly based on the membrane spanning body of the enzymes and is different between the individual SPPL-family members. The observations, that all SPPL2a and SPPL2b chimeras allow at least to some extent consecutive processing (Figs. 6 & 7) and that sites of consecutive cleavage are not altered by chimeric proteases (Fig. 8), further strengthens this hypothesis. Experimentally determined 3D-structures of presenilin in

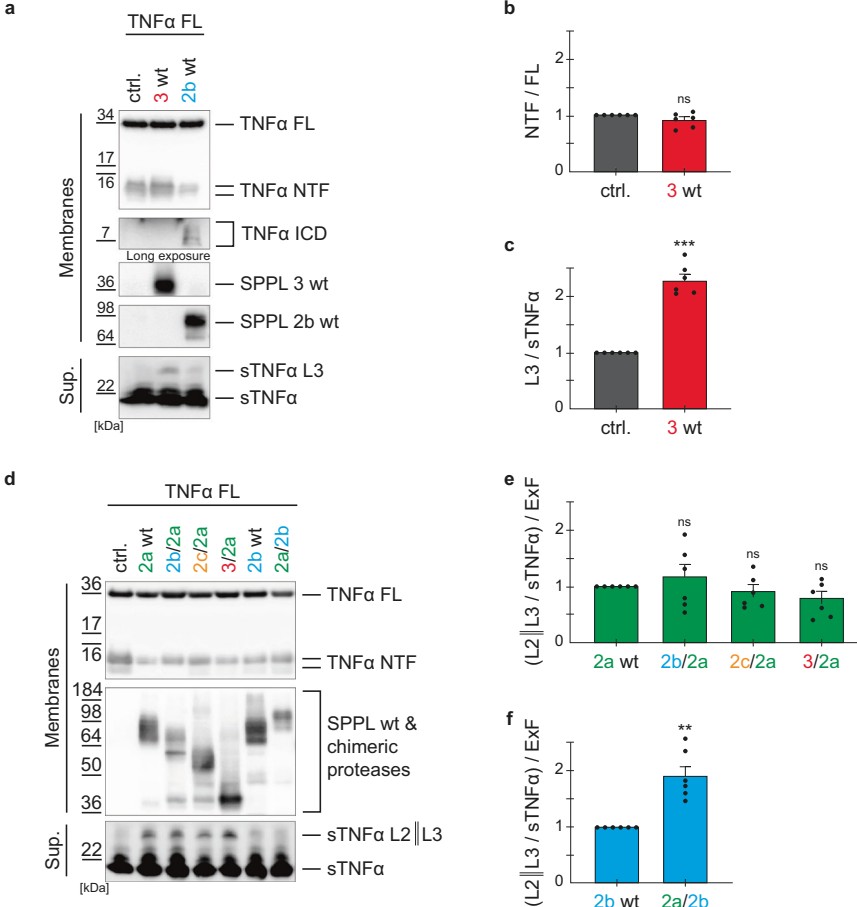

**Fig. 9 | Non-canonical shedding of TNFα by SPPL3 and chimeric SPPL2 proteases. a** DKO cells (ctrl.) and dKO cells stably overexpressing SPPL3 wt or SPPL2b wt were transiently transfected with full length TNFα (TNFα FL). Membrane bound TNFα species were analyzed as described in Fig. 5. SPPL3 wt was detected using the 7F9 antibody and SPPL2b wt using the 2G8 antibody. Soluble TNFα species were detected from Western Blot of conditioned media (supernatant, Sup.) with the monoclonal V5 antibody. SPPL2b expressing cells serve as a positive control for TNFα ICD production. **b** Densitometric quantification of non-cleaved TNFα NTF, as depicted in (**a**). Normalization to TNFα FL eliminated transfection variations. Results are depicted relative to the control cells (dKO). **c** Densitometric quantification of the SPPL3 non-canonical TNFα shedding product (sTNFα L3), as depicted in (**a**). Normalization to the ADAM-generated TNFα shedding product (sTNFα) eliminated transfection variations. Results are depicted relative to the control cells

(dKO). **d** The experiment was performed as described in Fig. 4 but includes the soluble TNFα fragments. **e, f** Densitometric quantification of non-canonical TNFα shedding products (sTNFα L2∥L3) generated by either the SPPL2a (**e**) or SPPL2b (**f**) chimeric proteases, as depicted in (**d**). Normalization to the ADAM-generated TNFα shedding product (sTNFα) eliminated transfection variations. Expression differences between chimeric and wt proteases are accounted for by division with the same ExF values as in Fig. 4 (ExF values of 4b, d equals those in (**e**), ExF values of 4c, e equal those in (**f**). Resulting values were all normalized to their corresponding wt protease sample. Mean + SEM, unpaired, two-tailed one sample t-tests of log-transformed ($\log_2$) values. ns not significant, **$p < 0.01$, ***$p < 0.001$ (**b**: p of 3 wt=0.156; **c**: p of 3 wt<0.0001; **e**: p of 2b/2a = 0.9163, p of 2c/2a = 0.269, p of 3/2a = 0.103; **f**: p of 2a/2b = 0.001), $n = 6$ independent experiments.

the γ-secretase complex, suggest the processive substrate cleavage is based on the architecture of the catalytic cleft and its surrounding in the membrane[30,31,58,59]. Together with our data, this may indicate that the membrane surrounding of the SPPL3 active cleft differs from that of the other family members.

To our surprise, the PA domain of SPPL2a and the short N-terminus of SPPL3 both improved SPPL2b mediated processing of TNFα NTF species generated from TNFα FL in vitro (Fig. 5b), but significantly impaired processing of TNFα NTF mimicking the major ADAM cleavage product under the same conditions (Fig. 10b). TNFα NTF species derived in vitro from TNFα FL comprise ADAM-generated membrane bound species but also those that are produced from the chimeric proteases and to a minor extent from the endogenously present SPPL3, while in the other assay condition, only one defined TNFα NTF species is present, which terminates at the major ADAM17 cleavage site[39] followed by a V5-tag. This indicates that length and/or quality of the substrate's luminal juxtamembrane domain significantly affect the interaction with the N-terminal domain of the protease. In the γ-secretase complex, the luminal juxtamembrane regions of the

substrates Notch and APP were shown by cryo-electron microscopy to interact with nicastrin[30,31]. Given that the N-terminal PA domains are also located on the luminal side, they had been suggested to play a similar role[22]. Our data further support this idea. Interestingly, combination of the N-termini of the two non-canonical sheddases with SPPL2b, which is not capable of non-canonical shedding[39], enhanced recognition of other, most likely shorter, membrane bound TNFα species (Fig. 5b) and allowed processive cleavage (Fig. 7a, c), which is intrinsic to SPPL2b. Also in intact cells, the SPPL2a N-terminal PA domain enhanced TNFα NTF turnover by SPPL2b (Fig. 4c) and allowed a small, but significant amount of non-canonical shedding compared to the background signal detected in SPPL2 wt expressing and control cells (Fig. 9d). Consistent with earlier data, even the overexpression of SPPL2b wt hardly shows non-canonical shedding activity[39]. Considering that the SPPL2a/2b chimera is less expressed compared to SPPL2b wt (Fig. 9d), it exhibits increased efficiency in TNFα non-canonical shedding (Fig. 9f), but clearly weaker efficiency than that of SPPL2a wt. Based on our data on SPPL3 acting as a non-canonical sheddase (Fig. 9a, c), the sTNFα L2∥3 background signal most likely results from

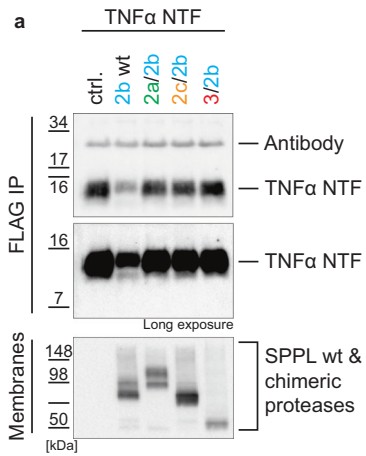

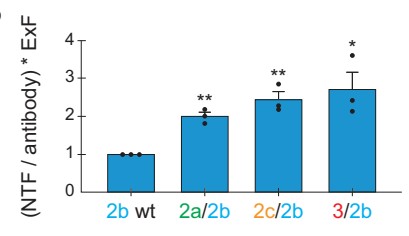

**Fig. 10 | In vitro cleavage of TNFα NTF by SPPL2b chimeric proteases. a** DKO cells were either untransfected (ctrl.) or transiently transfected with SPPL2b wt or the indicated chimeric proteases. In vitro cleavage of separately expressed TNFα NTF terminating at the major ADAM17-cleavage site followed by a V5 tag (TNFα NTF) was carried out and TNFα species were immunoprecipitated (FlagM2) and analyzed on Western Blot (polyclonal anti-Flag antibody). SPPL proteases were detected in solubilized membranes with an antibody against the HA-tag (3F10). **b** Densitometric quantification of non-cleaved TNFα NTF, as depicted in (**a**).

Normalization to the antibody signal eliminated experimental variations. Expression differences between chimeric and wt proteases are accounted for by multiplication with ExF. Resulting values were all normalized to the 2b wt protease sample. Mean + SEM, unpaired, two-tailed one sample t-tests of log-transformed (log$_2$) values. ns not significant, *$p < 0.05$, **$p < 0.01$ (p of 2a/2b = 0.006, p of 2c/2b = 0.0087, p of 3/2b = 0.0254), $n = 3$ independent experiments. The ExF is the mean ($n = 3$) of the ratio between the expression of the individual chimera and SPPL2b wt.

endogenous SPPL3 and has also been seen before[39]. This suggests that the ability of non-canonical shedding is to some extent determined by the N-terminal domain. However, since the SPPL2a/2b chimera did not achieve the same non-canonical shedding activity as SPPL2a wt (Fig. 9d), other determinants in the membrane spanning part or the C-terminus of the SPPL proteases must contribute to this capability.

Our data also do not fully exclude that the N-termini of the proteases cooperatively interact with other parts of the protease or foster the formation of homodimers or other multimers that support substrate specificity or selectivity. At least prior to shedding and release of sTNFα as homotrimeric cytokine[60], TNFα FL exists as trimer in the membrane. It is not clear whether the TNFα NTF remains trimeric in the membrane before processive cleavage through SPPL2a and SPPL2b. Recently, detergent purified NTF species of TNFα and CD74 were shown to exist as dimers and trimers up to a certain length[61]. This could imply that multiple SPPL2a and SPPL2b enzymes may localize around such substrates, each processing one substrate TM domain by unfolding it and forming hybrid β-sheets in the active site of each subunit. However, this is highly speculative and would require thorough structural biology experiments to elucidate. Nevertheless, the N-terminal domains of the SPPL2 subfamily are homologous to PA domains that are known to act in substrate recognition and dimerization processes[22]. One observation in our study that might support such cooperative interactions is the reduced processivity observed in the SPPL2b/2a chimera (Fig. 6a), which is not accompanied by a reduced turnover of TNFα NTF in intact cells (Fig. 4b, d). This might be explained by a positive effect of the SPPL2a PA domain on its own processivity, which is lost by replacement with the SPPL2b PA domain, or by a compensatory mechanism over time, since in intact cells time for substrate processing by SPPL2 is much longer than in isolated membranes (Fig. 6a, d).

Together, our data strongly support that the N-termini of SPPL proteases, in particular the PA domains of the SPPL2 subfamily, are involved in substrate recognition and that the processivity of the enzymes is mainly determined by their membrane spanning parts. SPPL2a and SPPL2b have a strong overlapping substrate spectrum compared to SPPL2c[22]. Consistently, the conserved surface patch between SPPL2a and SPPL2b absent in the comparison with SPPL2c (Fig. 1a), further strengthens our finding and provides a starting point for future analysis of substrate-enzyme interaction.

Yet, we did not observe all-or-nothing effects, demonstrating that the specific interplay between different parts of the enzymes, as well as of the

substrate, is crucial for efficient proteolysis by SPPL proteases. Since we have only analyzed recognition of TNFα by SPPL2-family members, we cannot exclude that other substrates interact differently with the enzymes. Thus, this study will only be the first indication on which protease intrinsic features are crucial for substrate recognition and processing, and the analysis of more enzyme substrate combinations, as well as the determination of experimentally derived 3D-strucures of substrates bound to the enzymes will be required to achieve a full mechanistic understanding of these important proteases.

## Methods
### Antibodies
For Western blot analysis, the following primary antibodies were used: Anti-FLAG M2® (1:1000, mouse mAb, Sigma-Aldrich, Darmstadt, Germany), anti-FLAG® (1:3000, rabbit, pAB, clone F7425, Sigma-Aldrich, Darmstadt, Germany), anti-V5 (1:2000, mouse mAb, clone R960-25, Invitrogen, Hennigsdorf, Germany), anti-Calnexin (1:1000, rabbit pAb, Enzo Life Sciences, Lörrach, Germany), and horseradish peroxidase-conjugated anti-HA (anti-HA-HRP: 1:3000, rat, mAb, clone 3F10, Roche, Mannheim, Germany), anti-SPPL3 and anti-SPPL2b monoclonal antibodies (1:10) have been described earlier[62,63]. Secondary antibodies used were: HRP-conjugated goat anti-mouse (1:10000), anti-rabbit (1:20000) (Promega, Walldorf, Germany) and anti-rat (1:3300, Merck, Darmstadt, Germany). For immunofluorescence stainings, the following primary antibodies were utilized: anti-BiP (1:1000, rabbit, clone ab21685 pAb, Abcam, Germany), anti-GM130 (1:500, mouse, mAB, clone 35, BD Bioscience, Heidelberg, Germany), anti-Lamp1 (1:100, rabbit, mAb, clone D2D11, Cell Signaling, Frankfurt am Main, Germany), and fluorescein-conjugated anti-HA (1:20, rat, mAb, clone 3F10, Roche, Mannheim, Germany). Alexa 555-conjugated secondary antibodies included goat anti-mouse and goat anti-rabbit (1:1000, Thermo Fisher, Bremen, Germany). For actin staining 1x phalloidin was used (Thermo Fisher, Bremen, Germany).

### cDNA, molecular cloning
The cDNAs encoding SPPL2a wt (NCBI Reference Sequence: NM_032802.3), SPPL2b wt (NCBI Reference Sequence: NM_152988.32), and SPPL3 wt (NCBI Reference Sequence: NM_139015.4) with a C-terminal HA-tag have been previously described[63]. TNFα FL wt (NCBI Reference Sequence: NM_000594.3) and TNFα NTF comprise N-terminal FLAG and C-terminal V5

tags and have been described earlier[39]. DNA constructs encoding chimeric proteases with a C-terminal HA-tag were designed using CLC software and purchased from IDT or Genscript (https://www.genscript.com/). They were subcloned into pcDNA4 TO myc-His A target vector (Invitrogen, Bremen, Germany) via Xho1 (Thermo Scientific, Bremen, Germany) and EcoR1 (New England Biolabs, Frankfurt am Main, Germany) restriction sites. Sequence verification was performed through Sanger sequencing, and all DNA sequences are available upon request, the full protein sequences of the chimeric proteases are depicted in Suppl. Figures 1 & 2.

## Cell culture and stable cell lines

T-Rex™-293 SPPL2a/b double knockout cells (dKO) have been previously described[39]. DKO cells (ctrl.) were cultured in Dulbecco's Modified Eagle Medium (DMEM) with GlutaMAX™ (Invitrogen, Hennigsdorf, Germany) supplemented with 10% fetal calf serum (Sigma), 1% penicillin/streptomycin (Gibco) and 5 µg/ml blasticidin (Invitrogen, Hennigsdorf, Germany). DKO cell lines stably expressing SPPL wt or chimeric proteases have been established using 8 µl Lipofectamine™ 2000 (Thermo Fisher, Bremen, Germany) and 20 µg of the respective cDNA following the manufacturer's instructions. 48 h post-transfection zeocin-driven selection was performed, either single cell clones or a pool of stable cell lines were established. Zeocin selection and cultivation of stable cell lines were performed in DMEM, high glucose with GlutaMAX™ (catalog number: 61965059, Thermo Fisher, Bremen, Germany) supplemented with 10% fetal calf serum (Sigma), 1% penicillin/streptomycin (Gibco), 5 µg/ml blasticidin (Invitrogen, Hennigsdorf, Germany) and 10 µg/ml zeocin (Invitrogen, Hennigsdorf, Germany). Due to the low protease expression of the SPPL2a/2b chimera, a single clone with high expression was selected. All other cell lines were pool stables. All SPPL wt and chimeric proteases contain a C-terminal HA-tag (Suppl. Figures 1 & 2). Protease expression was induced for up to 48 h prior to harvesting with 10 ng/ml (immunofluorescence stainings) or 1 µg/ml doxycycline (BD Biosciences, San Jose) in an otherwise antibiotic-free medium.

## Protein extraction and immunoblotting

Prior to experiments, cells were seeded on poly-L-lysine coated dishes in the appropriate media, followed by doxycycline induction 24 h later. To investigate TNFα cleavage, transient transfection of TNFα FL was performed 24 h after doxycycline induction using Lipofectamine™ 2000 (Thermo Fisher, Bremen, Germany) following the manufacturer's instructions. Cells were harvested on ice 48 h post-induction for protein isolation. For analysis of secreted TNFα cleavage products the conditioned media was collected, cleared for 20 min at 16200 g and 4 °C and mixed with 5x sample buffer (50% (v/v) glycerol, 7.5% (w/v) SDS, 7.5% (w/v) DTT, traces of bromophenol blue, dissolved in 4x upper Tris buffer).

To enrich membrane proteins, cells were lysed in ice-cold hypotonic buffer (10 mM Tris, 1 mM EDTA, 1 mM EGTA, pH 7.6), supplemented with 1:500 protease inhibitor mix (P1860, Sigma Aldrich, Darmstadt, Germany) and mechanically homogenized using a 23 G needle. Membranes were isolated at 4 °C through centrifugation for 5 min at 1200 g and for 45 min at 16200 g. Pellets were resuspended in basic buffer (40 mM Tris, 40 mM potassium acetate, 1.6 mM magnesium acetate, 100 mM sucrose, 0.8 mM DTT) and 2x sample buffer (10% (v/v) glycerol, 7.5% (w/v) SDS, 7.5% (w/v) DTT, traces of bromophenol blue, dissolved in 4x Tris (0.5 M Tris, 0.8% (w/v) SDS pH 6.8), was added.

To analyze protease expression, samples were incubated for 10 min at 65 °C and 750 rpm on a ThermoMixer® C (Eppendorf, Hamburg, Germany). Proteins were separated on a 12% SDS-PAGE gel, and protein transfer was conducted at 400 mA for 60 min onto a polyvinylidene fluoride (PVDF) membrane (PVDF; immobilon P transfer membrane, 0.45 mm pore width; Millipore). For analysis of TNFα species, samples were incubated for 5 min at 95 °C and 750 rpm on a ThermoMixer® C (Eppendorf, Hamburg, Germany). Separation of intracellular TNFα species was conducted on a modified Tris-Tricine gel[7] and protein transfer was performed with an activated gel (incubated for 2 min in water and 15 min in blotting

buffer) for 30 min at 400 mA onto a PVDF membrane. Soluble TNFα fragments were separated on a 15% SDS gel, and after western blot transfer onto a nitrocellulose membrane for 60 min at 400 mA, the membranes were boiled for 5 min in PBS. Protein detection was conducted using Pierce™ ECL Western Blotting Substrate (Thermo Fisher, Hennigsdorf, Germany) or Westar Antares (Cyanagen, Bologna, Italy) according to the manufacturer's instructions. Chemiluminescence detection was performed using the ChemiDoc imaging system (Bio-Rad), and the evaluation was carried out using Image Lab (ChemiDoc). Contrast and brightness were adjusted for visualization purposes.

## Time dependent TNFα cleavage

To investigate TNFα kinetics over time, membrane pellets were resuspended in basic buffer supplemented with 1:500 protease inhibitor mix (P1860, Sigma Aldrich, Darmstadt, Germany). Protein amount was determined using the Rapid Gold BCA Protein Assay Kit (Pierce, Darmstadt, Germany) and 40 µg total protein were incubated at 37 °C and 750 rpm on a ThermoMixer® C (Eppendorf, Hamburg, Germany) for the indicated time points before 2x sample buffer was added. Protein separation, Western Blotting and detection was carried out as described above.

## Immunoprecipitation and MALDI-TOF mass spectrometry

Membrane pellets were resuspended in 200 µl basic buffer supplemented with protease inhibitor mix (1:500) and incubated for 30 min at 37 °C, followed by the solubilization in basic buffer supplemented with 2% N-dodecyl-β-D-maltoside (DDM, Calbiochem, San Diego, United States) for 30 min on ice. To isolate TNFα peptides, samples were centrifuged for 20 min at 16000 g and 4 °C, and supernatants were immunoprecipitated using anti-Flag®M2 agarose affinity beads (Sigma-Aldrich, Darmstadt, Germany) for 2 h at room temperature. Beads were washed 3 times with washing buffer (0.14 M NaCl, 0.1% N-octylglucopyranoside, 10 mM Tris-HCl pH 7.6, 5 mM EDTA) and 2 times with ddH$_2$O. The α-cyano-4-hydroxycinnamic acid Matrix was mixed 1:1 with acetonitrile and 0.6% TFA, and 10 µl was added to the beads containing the peptides. Three times 0.4 µl (total 1.2 µl) of each sample was spotted on an MTP 384 ground steel target plate (Bruker Daltonik GmbH, Germany) and left to dry at room temperature. Mass spectra were recorded on a rapifleX MALDI Tissuetyper MALDI-TOF/TOF mass spectrometer (Bruker Daltonik GmbH) in the linear mode with external calibration.

## Immunofluorescence staining and microscopy

Cells were seeded on poly-L-lysine coverslips in the appropriate media, followed by doxycycline induction 24 h later. Cells were washed twice with PBS, and fixation was performed with 4% paraformaldehyde (PFA, Hi-Media, Modautal, Germany) in PBS for 20 min at room temperature. Cells were then washed three times with PBS and permeabilized using blocking buffer (5% (w/v) donkey serum (Merck, Darmstadt, Germany) in PBS) supplemented with 0.1% (v/v) Triton X-100 for 5 min at room temperature, followed by 1 h incubation in blocking buffer at room temperature. Cells were incubated with primary antibodies overnight at 4 °C, with secondary antibodies or phalloidin for 1 h at room temperature, followed by an incubation in fluorescein-conjugated anti-HA (3F10) antibody for 30 min at 4 °C. All antibodies and phalloidin were diluted in blocking buffer, and after each step, cells were washed six times using PBS. Before mounting, cells were washed additionally with ddH2O. Mounting was performed face-down onto a microscope slide using ProLong™ Gold mounting medium (refractive index (RI) of 1.46) with 4,6-Diamidin-2-phenylindol (DAPI) (Thermo Fisher, Bremen, Germany) for nuclear staining and dried at room temperature overnight. Images were captured using a Stellaris 5 Confocal Microscope (Leica), equipped with a 63x/1.30 GLYC HC PL APO CS2 objective with a numerical aperture (NA) of 1.3, 3x zoom, and 1 Airy Unit (AU). The acquisition settings included a scan speed of 100 Hz, a resolution of 1024 × 1024 pixels, and a line average of 4. For fluorochrome excitation and detection, DAPI was excited with a 405 nm laser (via UV Laser Diode 405), with an emission range of 425–504 nm, detected by the HyD S 1 SiPM

detector. Fluorescein was excited by a pulsed white light laser (485-685 nm) at 499 nm, with an emission range of 504–556 nm, detected by the HyD S 2 SiPM detector. Alexa Fluor 555 was excited by a pulsed white light laser (485-685 nm) at 553 nm, with an emission range of 557.9–729.9 nm, detected by the HyD S 3 SiPM detector. Images were processed using the LIGHTNING module. The intensity of the images was increased using LAS X Office 1.4.4.26810 (2023) software. Scale bars were added accordingly. The LUT used for image processing was linear, covering the full range of the data with 8-bit depth.

### In vitro Assay

The in vitro assay for TNFα NTF cleavage by SPPL2b wt has been described, previously[51]. In brief, dKO cells were transiently transfected 24 h post-seeding using either 5 µg of cDNA encoding TNFα FL, TNFα NTF, SPPL2b wt, SPPL2c/2b, SPPL3/2b or 10 µg cDNA of SPPL2a/2b. After 48 h of doxycycline (1 µg/ml) induction, membranes were isolated and solubilized in assay buffer containing 40 mM Tris (pH 7.8), 40 mM potassium acetate, 1.6 mM magnesium acetate, 100 mM sucrose, 5% (v/v) glycerol, 0.026% SDS, 1% (v/v) CHAPSO, 5 mM DTT, and protease inhibitor mix (1:500) (P1860, Sigma Aldrich, Darmstadt, Germany). Samples were incubated on ice for 1 h and then centrifuged (30 min at 100000 g, 4 °C). For the in vitro assay, the supernatants of control or protease expressing cells were mixed with supernatants from samples either expressing TNFα FL or TNFα NTF in assay buffer supplemented with 0.05% cholesterol and 1.5 mg/ml phosphatidylcholine. Anti-FLAG M2 antibody-coupled agarose beads (ANTI-FLAG® M2 Affinity Gel, Sigma-Aldrich, Darmstadt, Germany) were added, and samples were incubated while rotating for 2 h at 37 °C, followed by a pull down of TNFα NTF, TNFα FL and TNFα ICDs via centrifugation (3 min at 3000 g, 4 °C). Samples were washed twice with assay buffer containing a reduced CHAPSO concentration (0.5% CHAPSO), supplemented with SDS-sample buffer, boiled and separated on a modified Schägger gel[7]. For the detection of SPPL2b wt and SPPL2b chimeric proteases, the solubilized membranes were mixed with SDS-sample buffer, boiled and separated on a 12% gel. Western blot analysis was performed as described above.

### 3D conservation analyses

Sequences of human SPPL2a (uniprot-ID Q8TCT8), SPPL2b (uniprot-ID Q8TCT7), and SPPL2c (uniprot-ID Q8IUH8) were aligned with Clustal Omega[48,49] and conservation was mapped using Pymol[64] onto the AlphaFold[46,47] predicted surface of the N-terminal PA domain of SPPL2a. Residues were colored according to the Clustal Omega output file with identical residues shown in dark green, chemically closely related residues in light green, and chemically loosely related residues in yellow. The globular PA domains were depicted as solvent accessible surfaces. The analogous procedure was performed for a comparison only of human SPPL2a and human SPPL2b as well as for a cross-mammalian species analyses. For the later, next to the above human sequences the following sequences of *Sus scrofa*, *Mus musculus*, and *Sarcophilus harrisii* were used: SPPL2a - A0A4X1SWN6, Q9JJF9, G3WJY2; SPPL2b - A0A287BCL8, Q3TD49, G3VS63; SPPL2c - F1RRS5, A2A6C4, G3WW48.

### Statistics and reproducibility

Densitometric quantification of Western blots was performed using Image Lab software (Bio-Rad). Statistical analysis was carried out using Graph Pad Prism (Version 9.4.1), and Adobe Illustrator 2024 (Version 28) was used for graph and plot visualization. All experiments were performed with at least three biological replicates (cells seeded independently with at least 1 week apart). To account for loading and transfection variations within the biological experiments, fragments of interest were normalized to a suitable internal loading control. For TNFα ICD, the internal loading control was TNFα FL, and the sTNFα L2‖3 fragments were normalized to ADAM-produced sTNFα. Depending on the experiment, TNFα NTF was normalized to either TNFα FL (Fig. 4, Fig. 9, Suppl. Figure 3, and Suppl. Figure 4), calnexin (Fig. 6, Fig. 7, Suppl. Figure 5), or the ~25 kDa antibody light chain band (Figs. 5, 10).

To account for differences in protease expression, the expression of the individual chimeric proteases was determined in each biological replicate by quantifying the total protease amount relative to the total amount of its corresponding wt protease. To account for differences in protease expression, the expression factor (ExF) was introduced. The expression of the individual chimeric protease in each biological replicate was determined by quantifying the total protease amount (at time point 0) relative to the total amount of its corresponding wt protease. The mean of these individual expression ratios defined the ExF for the corresponding chimeric protease. To compare protease activities, the increase in product (TNFα ICD and TNFα L2) was divided by the ExF, while the remaining substrate (TNFα NTF) was multiplied by the ExF.

For quantification of time dependent TNFα NTF reduction, the remaining TNFα NTF at each time point was normalized to calnexin. The log-transformed ($log_2$) result was plotted against time (x = time, y = $log_2$(NTF/calnexin)). The slope of the resulting regression curve for the respective chimeric protease was divided by the ExF at time point 0 and normalized to the corresponding wt slope.

Throughout all experiments values are displayed relative to the corresponding wt or control (set to 1) and all statistical analyses were performed on log-transformed ($log_2$) values of the results. To test whether the average of each chimera differed significantly from the wt or control, an unpaired, two-tailed one-sample t-test was performed. A significance level of $p < 0.05$ (*$p < 0.05$, **$p < 0.01$, ***$p < 0.001$) was set, and samples are presented as mean ± standard error of the mean (SEM). The Shapiro-Wilk test was used to test for normal distribution.

### Reporting summary

Further information on research design is available in the Nature Portfolio Reporting Summary linked to this article.

### Data availability

All supplemental figures referred to in the main text as well as all uncropped images of Western Blot-based experiments shown in the main figures are available in the Supplementary Information linked to this article. The numerical source data for all graphs and charts depicted in the main and supplemental figures are available in the Supplementary Data 1 linked to this article. The raw data of the MALDI-TOF data in Fig. 8 and supplemental Fig. 6, as well as the sequence verifications of all chimeric enzymes are provided as Supplementary Data 2 linked to this article. Further information and requests for resources and reagents should be directed to and will be fulfilled by Regina Fluhrer (regina.fluhrer@med.uni-augsburg.de).

### Materials availability

Cell lines and cDNA constructs generated in this study will be made available upon reasonable request, but we may require a payment and/or a completed Materials Transfer Agreement if there is potential for commercial application.

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

## Acknowledgements

This work was supported by grants of the Deutsche Forschungsgemeinschaft to Regina Fluhrer (263531414 / FOR 2290; 254872893 / FL 635/2-3). The Confocal Microscope used in this study was funded by the Deutsche Forschungsgemeinschaft (507881424 / ITM-01). Regina Fluhrer received intramural funding by the University of Augsburg within the program "Forschungspotenzial besser nutzen!"; Sabine Hoeppner received intramural funding from the Faculty of Medicine at the University of Augsburg. We thank Shibojyoti Lahiri and the ZfP (Protein analysis Unit, BMC, Munich) for access to mass spectrometry and Rickmer Schulte and the StaBLab (Institute for Statistics, LMU Munich, Munich) for support in statistical analysis.

## Author contributions

K.S. carried out the in vitro assays shown in Figs. 5 and 10. M. H-K. and S.S. contributed to experiments shown in Figs. 6 and 7 and provided technical assistance. A.A.P. contributed to experiments shown in Fig. 8, M.J. and J.P. contributed to experiments shown in Figs. 4 and 9. H.H. and M.K. assisted with microscopic analysis of the immunofluorescence stainings shown in Fig. 3, C.S. performed all other experiments and contributed to writing of the manuscript. S.H. performed the prediction analysis shown in Fig. 1. S.H. and R.F. conceived the experiments, supervised the project, and wrote the manuscript with input from all authors.

## Funding

## Competing interests

The authors declare no competing interests regarding this study. They reported to have received fees as follows: R.F. and S.H. Elsevier Verlag.
