## [Transparent Peer Review file · Communications Biology]

The N-terminal PA domains of Signal-Peptide-Peptidase-like 2 (SPPL2) proteases impact on TNF α cleavage

Corresponding Author: Professor Regina Fluhrer

Version 0:

Reviewer comments:

Reviewer #1

(Remarks to the Author)

This is a very comprehensive study that seeks answers for the interesting substrate selectivity displayed by members of the signal peptidase like protease family. The data are solid and extensive. The prose is sharp. Control experiments are clearly described and appropriate. Statistics are well described, lending confidence in robustness of the study. I congratulate the authors on a very thorough presentation. There is very little to find fault with. Just a couple of minor points to consider.

Minor Points

- 1, Line 84. Please give the official EC number for this family.
2. The introduction, while being very thorough, is a bit long. If there was a reason to shorten it this could be done by abbreviating lines 173-191 since they are a recap of the methods, results and conclusions.
3. If space is a concern than Fig 1 could be eliminated since it is devoid of data and the concept is adequately described in the text.

Reviewer #3

(Remarks to the Author)

The authors investigate the importance of the protease associated (PA) domain of SPPL2a/b/c for substrate selectivity and catalytic efficiency of SPPL2a/2b/2c/3 against TNF. They employ AlphaFold predictions and evolutionary conservation analyses of the PA domains. This shows that the PA domains of SPPL2a/b/c may have different interaction surfaces, which suggests different regulatory properties. The authors go on making PA swaps between the individual enzymes and analyse the in-cell and in vitro cleavage properties of the resulting chimeric enzymes against TNF, the natural substrate of SPPL2a/b. The in-cell experiments are supported by localization analysis of the PA domain chimeras. They first establish, with SPPL2a and 2b, that protease expression levels correlate with TNF cleavage, which enables them to compare the activities of the chimeric enzymes taking into account their varying expression levels. After careful analysis they find that the PA domain is, akin to the nicastrin subunit of gamma secretase, involved in recognizing fine features of the ectodomain of the substrate (TNF in this case), while the processivity is dependent only on the transmembrane core of SPPLs. Overall, this is a very solid and novel piece of work, well-designed and controlled despite being technically challenging. These authors correctly exploit their in vitro assay for SPPL activity to measure kinetics. This in itself is state of the art in the field. The work should incite wide interest in the intramembrane protease community.

I have the following minor points and questions:

1. Western blots of SPPL variants are used for normalization of the cleavage data to expression levels. Since the enzymes are often glycosylated and form smeared bands, the normalization may be more accurate if deglycosylated samples were used for the western blots (such as those in Fig3B).

2. Section beginning on L219: it is not certain that substrate processing extent in cells should be linearly dependent on the protease levels. While this is true in vitro, the situation in cells is influenced by trafficking, lifetimes of the proteins, etc. I would therefore suggest to soften the description to "is proportional to the protease expression level" or similar. The validation of linearity presented in FigS3, 4 is made from two datapoints, two expression levels. Strictly speaking, this is not sufficient for inferring linearity of a relationship. Also, Fig.S3 and S4 are named "Mathematical relation between....". There is not much mathematics in there. I suggest renaming the titles to perhaps more appropriate "Quantitative relationship between..."

3. I wonder whether the transmembrane form of TNF that is a substrate for SPPLs is the monomeric species, or the trimeric form may be cleaved too.

4. The authors used TNF as a model substrate, but mention that the substrate spectrum is wider for SPPLs. Are the PA domain effects significant also for some other substrate?

5. There are occasional typos, such as "extend" that should instead read "extent", or "then" should be "than" on L576, and commas deserve revision throughout.

6. Fig. 12b panel is a duplication of Fig. 2a. While it is clear why Fig. 12 is shown, can these panel be merged into Fig. 2 instead? Can this be accommodated into the logical narrative of the paper?

Version 1:

Reviewer comments:

Reviewer #1

(Remarks to the Author)

the authors have provided an adequate response to my comments.

Reviewer #3

(Remarks to the Author)

The authors have answered all questions and reacted to the reviewers' comments, improving the manuscript. I support its publication in this journal.

Point-to-point responses to the reviewers' comments

We are very delighted that both reviewers consider our data solid and state of the art, and the manuscript important and of interest to the readers of Communications Biology. We thank them for their time, effort and helpful comments. Their comments and suggestions have improved the quality of our manuscript.

Reviewer #1

We appreciate that the reviewer values our study as comprehensive, solid and extensive and thank him/her for time and effort put in reviewing our manuscript.

1.

Comment: Line 84. Please give the official EC number for this family.

Response: To our knowledge, the intramembrane aspartyl proteases in particular presenilins and SPPL proteases so far have not been assigned separate EC numbers. Only SPP itself has recently been assigned the preliminary BRENDA-supplied EC number EC 3.4.23.B24. In the MEROPS database all representatives of the intramembrane aspartyl protease family are listed as Peptidase family A22, with presenilins forming subfamily A22A and the SPP/SPPL family subfamily A22B.

Revision: We added the preliminary EC number of SPP in line 88 and the EC 3.4.23 as general identifier of aspartyl proteases in line 90.

2.

Comment: The introduction, while being very thorough, is a bit long. If there was a reason to shorten it this could be done by abbreviating lines 173-191 since they are a recap of the methods, results and conclusions.

Response: We agree that the summary at the end of the introduction is a bit lengthy. However, the guidelines of the Journal explicitly request a "final paragraph that should be a brief summary of the major results and conclusions".

Revision: We have significantly shortened this paragraph (new lines 178 to 195) in the revised version but kept a few summarizing sentences to meet the Journal's guideline.

3.

Comment: If space is a concern than Fig 1 could be eliminated since it is devoid of data and the concept is adequately described in the text.

Response: Thanks for this comment. We initially thought to provide this figure to allow easy understanding of the concept of TNF α proteolysis and the labeling of the following figures, in particular for readers not familiar with this process. However, we realized that we have exceeded the maximum number of figures allowed.

Revision: We have removed figure 1 and adapted numbering of all figures accordingly. Text adaptations see blue marks in lines 213 to 217.

Reviewer #3

We are delighted that the reviewer recognizes our work as solid, novel and well designed, as well as of interest for the readers of Communications Biology. We thank him/her for their time and effort put in reviewing our manuscript.

1.

Comment: Western blots of SPPL variants are used for normalization of the cleavage data to expression levels. Since the enzymes are often glycosylated and form smeared bands, the normalization may be more accurate if deglycosylated samples were used for the western blots (such as those in Fig3B).

Response: We thank the reviewer for this valuable comment with which we fully agree. When establishing and validating our methods of quantification, we also had identified the multiple, glycosylation dependent SPPL species, which appear as smear on Western Blot, as challenging. We had therefore verified upfront, that the glycosylation status in the gel used for quantification does not impact on the relative amounts of proteases. As also suggested by the reviewer, we de-deglycosylated the samples using N-glycosidase F before separation on Western Blot (see figure a below). To judge whether the amount of enzyme differs between glycosylated and deglycosylated samples, we quantified the two samples relative to calnexin including all species (smear) larger than the respective monomeric form of the enzyme. We then analyzed the deglycosylated samples relative to the respective glycosylated sample that was set to 1. This resulted in very similar enzyme amounts in both conditions (see Figure b below). We had thus concluded that our quantification approach is valid independent of the glycosylation status in the Western Blot. On the other hand, efficiency of the additional deglycosylation step would probably slightly vary between different experiments leading to an additional source of error and impacting on the accuracy of the results. Carefully weighing these options and considering that even when using an excess amount of N-glycosidase F we could not fully abolish the smear (see Figure a below), we decided that it would be more accurate to use the glycosylated protease version for quantification.

b

	n1	n2
2a	1	1
2a PNGase F	0,92925626	0,98063903
2b/2a	1	1
2b/2a PNGase F	1,17711027	1,07127205
2c/2a	1	1
2c/2a PNGase F	1,00021314	1,13105491
3/2a	1	1
3/2a PNGase F	0,80499139	1,10593814
2b	1	1
2b PNGase F	1,06623364	1,29394034
2a/2b	1	1
2a/2b PNGase F	0,75387998	1,16409507

Impact of N-glycosylation on quantifiable protease amount. a) Solubilized membranes from dKO cells overexpressing wild-type (wt) or chimeric proteases were treated overnight at 37°C with (+) PNGase F to remove N-linked glycosylation. Buffer-only treated samples (-) served as control. Samples were analyzed by western blot using an anti-HA antibody (3F10). Samples marked by double lines, were EndoH treated and are not relevant to the quantification in b). b) Densitometric analysis of glycosylated and deglycosylated proteases as depicted in (a). Samples were normalized to Calnexin as loading control and PNGase treated samples are depicted relative to their untreated control (PNGase F -). *Note: Results indicate comparable protease levels, irrespective of glycosylation status (n=2).*

Revision: Based on this observation, we decided not to include an additional step in our quantifications and use the glycosylated samples for determination of the protease amount. We consequently included all species larger than the full-length monomeric form of the enzyme in every quantification.

2.

Comment: Section beginning on L219: it is not certain that substrate processing extent in cells should be linearly dependent on the protease levels. While this is true in vitro, the situation in cells is influenced by trafficking, lifetimes of the proteins, etc. I would therefore suggest to soften the description to "is proportional to the protease expression level" or similar. The validation of linearity presented in FigS3, 4 is made from two datapoints, two expression levels. Strictly speaking, this is not sufficient for inferring linearity of a relationship. Also, Fig.S3 and S4 are named "Mathematical relation between...". There is

not much mathematics in there. I suggest renaming the titles to perhaps more appropriate "Quantitative relationship between..."

Response: We agree with the reviewer and apologize for this inaccuracy.

Revision: We replaced the terms "linear correlation" and "mathematical relation" in the manuscript, in the figures as well as in the supplement as suggested by the reviewer. Changes in the text are marked in blue starting at line 232.

3.

Comment: I wonder whether the transmembrane form of TNF that is a substrate for SPPLs is the monomeric species, or the trimeric form may be cleaved too.

Response: The question of TNF α 's trimerization during proteolysis is indeed intriguing. We are convinced that the actual cleavage has to happen in an at least partly monomeric form for the following reasons: Cryo-EM studies of presenilin, a close SPPL homolog, in complex with its substrates Notch and APP, show that its active site accommodates a single substrate TMD in a narrow channel, bound mainly by van-der-Waals forces from three surrounding helices (Yang et al., Nature 2019; Zhou et al., Science 2019) (Fig. 2a). The active site of this experimentally determined structure doesn't have space to accommodate an additional substrate TMD. SPPL proteases, predicted to have a very similar structure to presenilin (Hoeppner et al., 2022), also have 9 TMDs with two harboring their active site catalytic aspartates. It was shown that the substrate positioning in the active site of presenilin happens via formation of a hybrid beta-sheet between the substrate and the protease (Yang et al., Nature 2019; Zhou et al., Science 2019) (Fig. 2b). The residues involved in this beta-sheet are highly conserved between presenilins and SPPLs. It is therefore highly likely, that SPPL proteases bind and unfold their substrates in a very similar manner before cleavage. An intermediate confidence model that we calculated using AlphaFold3 supports this assumption (Fig. 2b).

Figure 2a: Cryo-EM structure of PS1 with APP; b: predicted structure of SPPL2b with TNF α ; enzymes are shown in blue, substrate peptides in pink

However, recently TNF α and other SPPL2a/b substrates were identified as so-called THOMAS proteins (*Thiolate-Heme Oligomeric type II Trans-Membrane proteins with Heme binding mode Adjusted by SPPL2a/b*) that can bind and sense heme (Kupke et al., 2020, Communications Biology). In these studies, TNF α NTF, purified from *E. coli*, was shown to form - *in vitro* and in detergent - stable trimers coordinating a heme group at the cytoplasmic site, even without the extracellular trimeric cytokine domain. The authors suggest that SPPL2a/b cleavage may alter the heme coordination and the oligomeric

state of TNF α NTF. This indeed raises the question how such an oligomeric state would interact with the SPPL2a/b active site if it would also be present *in vivo* in intact membranes. To structurally combine both findings, the C-terminal part of the TMD and extracellular juxtamembrane region of TNF α NTF would have to unfold into the active site, while the N-terminal TMD and cytoplasmic juxtamembrane region still maintain their oligomeric state. While this is well conceivable, it can in our opinion only be addressed with a thorough structural biology study that exceeds the scope of this study by far.

Revision: We have included our analysis of the current literature knowledge on the oligomeric state of TNF α into the discussion to highlight this open question in the field. Changes are marked in blue starting at line 592.

4.

Comment: The authors used TNF as a model substrate but mention that the substrate spectrum is wider for SPPLs. Are the PA domain effects significant also for some other substrate?

Response: Knowledge on cleavage characteristics for different SPPL2 substrates is still very limited and detection of cleavage products is highly variable and does not work as well as for TNF α .

To, nonetheless, get a first idea on this valid reviewer question, we co-expressed Bri2 with those chimeras that showed correct subcellular localization. However, the ICD species produced from the different chimeras depict different stability, which is most likely due to changes in consecutive cleavage as observed for TNF α . However, consecutive turnover of Bri2 ICD has not yet been detected and cleavage sites are also not established as the respective fragments are not well detected in MALDI-TOF mass-spectrometry. This, so far, makes it impossible to correctly interpret the data. In the future, a Bri2-based *in vitro* assay could answer this question. However, an *in vitro* assay exists so far only for TNF α . Thus, thoroughly answering this reviewer question would require establishment of new assays and, thus, comprise a completely new study and exceed the aim and feasibility of this work.

Revision: Based on this, we carefully checked our manuscript text to not overinterpret data and only draw conclusions on TNF α processing and avoid any generalization. In addition, we close the Discussion with the note that we have only analyzed recognition of TNF α by SPPL2-family members and provide an outlook that other enzyme-substrate combinations need to be investigated in the future (see lines 618-620).

5.

Comment: There are occasional typos, such as "extend" that should instead read "extent", or "then" should be "than" on L576, and commas deserve revision throughout.

Response: We apologize for any incorrectness.

Revision: We have corrected all mistakes mentioned by the reviewer and have proofread the whole manuscript. All changes and corrections in the manuscript have been marked in blue.

6.

Comment: Fig. 12b panel is a duplication of Fig. 2a. While it is clear why Fig. 12 is shown, can these panel be merged into Fig. 2 instead? Can this be accommodated into the logical narrative of the paper?

Response: We apologize that we have exceeded the maximum number of figures allowed.

Revision: As suggested by the reviewer we removed Figure 12 and merged the content with Figure 2, now new Figure 1 in the resubmitted version. Text and figure numbers have been adapted. Changes in the manuscript are marked in blue, in particular lines 213 to 217.